# MORE CENTRALIZED TRAINING, STILL DECENTRALIZED EXECUTION: MULTI-AGENT CONDITIONAL POLICY FACTORIZATION

**Jiangxing Wang**
School of Computer Science
Peking University
jiangxiw@stu.pku.edu.cn

**Deheng Ye**
Tencent Inc.
dericye@tencent.com

**Zongqing Lu**[†]
School of Computer Science
Peking University
zongqing.lu@pku.edu.cn

## ABSTRACT

In cooperative multi-agent reinforcement learning (MARL), combining value decomposition with actor-critic enables agents to learn stochastic policies, which are more suitable for the partially observable environment. Given the goal of learning local policies that enable decentralized execution, agents are commonly assumed to be independent of each other, even in centralized training. However, such an assumption may prohibit agents from learning the optimal joint policy. To address this problem, we explicitly take the dependency among agents into centralized training. Although this leads to the optimal joint policy, it may not be factorized for decentralized execution. Nevertheless, we theoretically show that from such a joint policy, we can always derive another joint policy that achieves the same optimality but can be factorized for decentralized execution. To this end, we propose *multi-agent conditional policy factorization* (MACPF), which takes more centralized training but still enables decentralized execution. We empirically verify MACPF in various cooperative MARL tasks and demonstrate that MACPF achieves better performance or faster convergence than baselines. Our code is available at https://github.com/PKU-RL/FOP-DMAC-MACPF.

## 1 INTRODUCTION

The cooperative multi-agent reinforcement learning (MARL) problem has attracted the attention of many researchers as it is a well-abstracted model for many real-world problems, such as traffic signal control (Wang et al., 2021a) and autonomous warehouse (Zhou et al., 2021). In a cooperative MARL problem, we aim to train a group of agents that can cooperate to achieve a common goal. Such a common goal is often defined by a global reward function that is shared among all agents. If centralized control is allowed, such a problem can be viewed as a single-agent reinforcement learning problem with an enormous action space. Based on this intuition, Kraemer & Banerjee (2016) proposed the centralized training with decentralized execution (CTDE) framework to overcome the non-stationarity of MARL. In the CTDE framework, a centralized value function is learned to guide the update of each agent's local policy, which enables decentralized execution.

With a centralized value function, there are different ways to guide the learning of the local policy of each agent. One line of research, called value decomposition (Sunehag et al., 2018), obtains local policy by factorizing this centralized value function into the utility function of each agent. In order to ensure that the update of local policies can indeed bring the improvement of joint policy, Individual-Global-Max (IGM) is introduced to guarantee the consistency between joint and local policies. Based on the different interpretations of IGM, various MARL algorithms have been proposed, such as VDN (Sunehag et al., 2018), QMIX (Rashid et al., 2018), QTRAN (Son et al., 2019), and QPLEX (Wang et al., 2020a). IGM only specifies the relationship between optimal local actions and optimal joint action, which is often used to learn deterministic policies. In order to learn stochastic policies, which are more suitable for the partially observable environment, recent studies (Su et al., 2021; Wang et al., 2020b; Zhang et al., 2021; Su & Lu, 2022) combine the idea of

---

[†]Corresponding Author

value decomposition with actor-critic. While most of these decomposed actor-critic methods do not guarantee optimality, FOP (Zhang et al., 2021) introduces Individual-Global-Optimal (IGO) for the optimal joint policy learning in terms of maximum-entropy objective and derives the corresponding way of value decomposition. It is proved that factorized local policies of FOP converge to the global optimum, given that IGO is satisfied.

The essence of IGO is for all agents to be independent of each other during both training and execution. However, we find this requirement dramatically reduces the expressiveness of the joint policy, making the learning algorithm fail to converge to the global optimal joint policy, even in some simple scenarios. As centralized training is allowed, a natural way to address this issue is to factorize the joint policy based on the chain rule (Schum, 2001), such that the dependency among agents' policies is explicitly considered, and the full expressiveness of the joint policy can be achieved. By incorporating such a joint policy factorization into the soft policy iteration (Haarnoja et al., 2018), we can obtain an optimal joint policy without the IGO condition. Though optimal, a joint policy induced by such a learning method may not be decomposed into independent local policies, thus decentralized execution is not fulfilled, which is the limitation of many previous works that consider dependency among agents (Bertsekas, 2019; Fu et al., 2022).

To fulfill decentralized execution, we first theoretically show that for such a *dependent* joint policy, there always exists another *independent* joint policy that achieves the same expected return but can be decomposed into independent local policies. To learn the optimal joint policy while preserving decentralized execution, we propose *multi-agent conditional policy factorization* (MACPF), where we represent the dependent local policy by combining an independent local policy and a dependency policy correction. The dependent local policies factorize the optimal joint policy, while the independent local policies constitute their independent counterpart that enables decentralized execution. We evaluate MACPF in several tasks, including matrix game (Rashid et al., 2020), SMAC (Samvelyan et al., 2019), and MPE (Lowe et al., 2017). Empirically, MACPF consistently outperforms its base method, *i.e.*, FOP, and achieves better performance or faster convergence than other baselines. By ablation, we verify that the independent local policies can indeed obtain the same level of performance as the dependent local policies.

## 2 PRELIMINARIES

### 2.1 MULTI-AGENT MARKOV DECISION PROCESS

In cooperative MARL, we often formulate the problem as a multi-agent Markov decision process (MDP) (Boutilier, 1996). A multi-agent MDP can be defined by a tuple $\langle I, S, A, P, r, \gamma, N \rangle$. $N$ is the number of agents, $I = \{1, 2 \ldots, N\}$ is the set of agents, $S$ is the set of states, and $A = A_1 \times \cdots \times A_N$ is the joint action space, where $A_i$ is the individual action space for each agent $i$. For the rigorousness of proof, we assume full observability such that at each state $s \in S$, each agent $i$ receives state $s$, chooses an action $a_i \in A_i$, and all actions form a joint action $\boldsymbol{a} \in A$. The state transitions to the next state $s'$ upon $\boldsymbol{a}$ according to the transition function $P(s'|s, \boldsymbol{a}) : S \times A \times S \rightarrow [0, 1]$, and all agents receive a shared reward $r(s, \boldsymbol{a}) : S \times A \rightarrow \mathbb{R}$. The objective is to learn a local policy $\pi_i(a_i|\mathrm{s})$ for each agent such that they can cooperate to maximize the expected cumulative discounted return, $\mathbb{E}[\sum_{t=0}^{\infty} \gamma^t r_t]$, where $\gamma \in [0, 1)$ is the discount factor. In CTDE, from a centralized perspective, a group of local policies can be viewed as a joint policy $\pi_{\mathrm{jt}}(\boldsymbol{a}|\mathrm{s})$. For this joint policy, we can define the joint state-action value function $Q_{\mathrm{jt}}(\mathrm{s}_t, \boldsymbol{a}_t) = \mathbb{E}_{\mathrm{s}_{t+1:\infty}, \boldsymbol{a}_{t+1:\infty}}[\sum_{k=0}^{\infty} \gamma^t r_{t+k}|\mathrm{s}_t, \boldsymbol{a}_t]$. Note that although we assume full observability for the rigorousness of proof, we use the trajectory of each agent $\tau_i \in \mathcal{T}_i : (Y \times A_i)^*$ to replace state $s$ as its policy input to settle the partial observability in practice, where $Y$ is the observation space.

### 2.2 FOP

FOP (Zhang et al., 2021) is one of the state-of-the-art CTDE methods for cooperative MARL, which extends value decomposition to learning stochastic policy. In FOP, the joint policy is decomposed into independent local policies based on Individual-Global-Optimal (IGO), which can be stated as:

$$\pi_{\mathrm{jt}}(\boldsymbol{a}|\mathrm{s}) = \prod_{i=1}^{N} \pi_i(a_i|\mathrm{s}). \tag{1}$$

As all policies are learned by maximum-entropy RL (Haarnoja et al., 2018), *i.e.*, $\pi_i(a_i|\mathrm{s}) = \exp(\frac{1}{\alpha_i}(Q_i(\mathrm{s}, a_i) - V_i(\mathrm{s})))$, IGO immediately implies a specific way of value decomposition:

$$Q_{\mathrm{jt}}(\mathrm{s}, \boldsymbol{a}) = \sum_{i=1}^{N} \frac{\alpha}{\alpha_i} [Q_i(\mathrm{s}, a_i) - V_i(\mathrm{s})] + V_{\mathrm{jt}}(\mathrm{s}). \tag{2}$$

Unlike IGM, which is used to learn deterministic local policies and naturally avoids the dependency of agents, IGO assumes agents are independent of each other in both training and execution. Although IGO advances FOP to learn stochastic policies, such an assumption can be problematic even in some simple scenarios and prevent learning the optimal joint policy.

## 2.3 PROBLEMATIC IGO

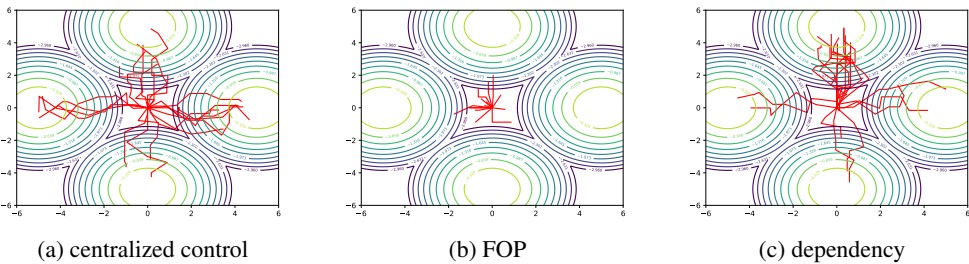

| (a) centralized control | (b) FOP | (c) dependency |

Figure 1: Sampled trajectories from the learned policy: (a) centralized control; (b) FOP, where IGO is assumed; (c) considering dependency during training.

As stated in soft Q-learning (Haarnoja et al., 2018), one goal of maximum-entropy RL is to learn an optimal maximum-entropy policy that captures multiple modes of near-optimal behavior. Since FOP can be seen as the extension of maximum-entropy RL in multi-agent settings, it is natural to assume that FOP can also learn a multi-modal joint policy in multi-agent settings. However, as shown in the following example, such a desired property of maximum-entropy RL is not inherited in FOP due to the IGO condition.

We extend the single-agent multi-goal environment used in soft Q-learning (Haarnoja et al., 2018) to its multi-agent variant to illustrate the problem of IGO. In this environment, we want to control a 2D point mass to reach one of four symmetrically placed goals, as illustrated in Figure 1. The reward is defined as a mixture of Gaussians, with means placed at the goal positions. Unlike the original environment, this 2D point mass is now jointly controlled by two agents, and it can only move when these two agents select the same moving direction; otherwise, it will stay where it is. As shown in Figure 1a, when centralized control is allowed, multi-agent training degenerates to single-agent training, and the desired multi-modal policy can be learned. However, as shown in Figure 1b, FOP struggles to learn any meaningful joint policy for the multi-agent setting. One possible explanation is that, since IGO is assumed in FOP, the local policy of each agent is always independent of each other during training, and the expressiveness of joint policy is dramatically reduced. Therefore, when two agents have to coordinate to make decisions, they may fail to reach an agreement and eventually behave in a less meaningful way due to the limited expressiveness of joint policy. To solve this problem, we propose to consider dependency among agents in MARL algorithms to enrich the expressiveness of joint policy. As shown in Figure 1c, the learned joint policy can once again capture multiple modes of near-optimal behavior when the dependency is considered. Details of this algorithm will be discussed in the next section.

## 3 METHOD

To overcome the aforementioned problem of IGO, we propose ***multi-agent conditional policy factorization (MACPF)***. In MACPF, we introduce dependency among agents during centralized training to ensure the optimality of the joint policy without the need for IGO. This joint policy consists of *dependent* local policies, which take the actions of other agents as input, and we use this joint policy as the behavior policy to interact with the environment during training. In order to fulfill

decentralized execution, *independent* local policies are obtained from these dependent local policies such that the joint policy resulting from these *independent* local policies is equivalent to the behavior policy in terms of expected return.

### 3.1 CONDITIONAL FACTORIZED SOFT POLICY ITERATION

Like FOP, we also use maximum-entropy RL (Ziebart, 2010) to bridge policy and state-action value function for each agent. Additionally, it will also be used to introduce dependency among agents. For each local policy, we take the actions of other agents as its input and define it as follows:

$$\pi_i(a_i|\mathrm{s}, a_{<i}) = \exp(\frac{1}{\alpha_i}(Q_i(\mathrm{s}, a_{<i}, a_i) - V_i(\mathrm{s}, a_{<i}))) \tag{3}$$

$$V_i(\mathrm{s}, a_{<i}) := \alpha_i \sum_{a_i} \exp(\frac{1}{\alpha_i} Q_i(\mathrm{s}, a_{<i}, a_i)), \tag{4}$$

where $a_{<i}$ represents the joint action of all agents whose indices are smaller than agent $i$. We then can get the relationship between the joint policy and local policies based on the chain rule factorization of joint probability:

$$\pi_{\mathrm{jt}}(\boldsymbol{a}|\mathrm{s}) = \prod_{i=1}^{N} \pi_i(a_i|\mathrm{s}, a_{<i}). \tag{5}$$

The full expressiveness of the joint policy can be guaranteed by (5) as it is no longer restricted by the IGO condition. From (5), together with $\pi_{\mathrm{jt}}(\boldsymbol{a}|\mathrm{s}) = \exp(\frac{1}{\alpha}(Q_{\mathrm{jt}}(\mathrm{s}, \boldsymbol{a}) - V_{\mathrm{jt}}(\mathrm{s})))$, we have the $Q_{\mathrm{jt}}$ factorization as:

$$Q_{\mathrm{jt}}(\mathrm{s}, \boldsymbol{a}) = \sum_{i=1}^{N} \frac{\alpha}{\alpha_i} [Q_i(\mathrm{s}, a_{<i}, a_i) - V_i(\mathrm{s}, a_{<i})] + V_{\mathrm{jt}}(\mathrm{s}). \tag{6}$$

Note that in maximum-entropy RL, we can easily compute $V$ by $Q$. From (6), we introduce *conditional factorized soft policy iteration* and prove its convergence to the optimal joint policy in the following theorem.

**Theorem 1** (**Conditional Factorized Soft Policy Iteration**). *For any joint policy $\pi_{\mathrm{jt}}$, if we repeatedly apply joint soft policy evaluation and individual conditional soft policy improvement from $\pi_i \in \Pi_i$. Then the joint policy $\pi_{\mathrm{jt}}(\boldsymbol{a}|\mathrm{s}) = \prod_{i=1}^{N} \pi_i(a_i|\mathrm{s}, a_{<i})$ converges to $\pi_{\mathrm{jt}}^*$, such that $Q_{\mathrm{jt}}^{\pi_{\mathrm{jt}}^*}(\mathrm{s}, \boldsymbol{a}) \geq Q_{\mathrm{jt}}^{\pi_{\mathrm{jt}}}(\mathrm{s}, \boldsymbol{a})$ for all $\pi_{\mathrm{jt}}$, assuming $|A| < \infty$.*

*Proof.* See Appendix A. □

### 3.2 *Independent* JOINT POLICY

Using the conditional factorized soft policy iteration, we are able to get the optimal joint policy. However, such a joint policy requires dependent local policies, which are incapable of decentralized execution. To fulfill decentralized execution, we have to obtain independent local policies.

| $a_1$ \ $a_2$ | $A$ | $B$ |
|---|---|---|
| $A$ | 0.5 | 0 |
| $B$ | 0 | 0.5 |

(a) dependent joint policy $\pi_{\mathrm{jt}}^{\mathrm{dep}}$

| $a_1$ \ $a_2$ | $A$ | $B$ |
|---|---|---|
| $A$ | 1 | 0 |
| $B$ | 0 | 0 |

(b) independent joint policy $\pi_{\mathrm{jt}}^{\mathrm{ind}}$

| $a_1$ \ $a_2$ | $A$ | $B$ |
|---|---|---|
| $A$ | −0.5 | 0 |
| $B$ | 0 | 0.5 |

(c) dependency correction $b_{\mathrm{jt}}^{\mathrm{dep}}$

Figure 2: A dependent joint policy and its independent counterpart

Consider the joint policy shown in Figure 2a. This joint policy, called *dependent joint policy $\pi_{\mathrm{jt}}^{\mathrm{dep}}$*, involves dependency among agents and thus cannot be factorized into two independent local policies. However, one may notice that this policy can be decomposed as the combination of an *independent joint policy $\pi_{\mathrm{jt}}^{\mathrm{ind}}$* that involves no dependency among agents, as shown in Figure 2b, and a dependency

policy correction $b_{\text{jt}}^{\text{dep}}$, as shown in Figure 2c. More importantly, since we use the Boltzmann distribution of joint Q-values as the joint policy, the equivalence of probabilities of two joint actions also indicates that their joint Q-values are the same,

$$\pi_{\text{jt}}^{\text{dep}}(A, A) = \pi_{\text{jt}}^{\text{dep}}(B, B) \Rightarrow Q_{\text{jt}}(A, A) = Q_{\text{jt}}(B, B). \tag{7}$$

Therefore, in Table 2, the expected return of the independent joint policy $\pi_{\text{jt}}^{\text{ind}}$ will be the same as the dependent joint policy $\pi_{\text{jt}}^{\text{dep}}$,

$$\mathbb{E}_{\pi_{\text{jt}}^{\text{dep}}}[Q_{\text{jt}}] = \pi_{\text{jt}}^{\text{dep}}(A, A) * Q_{\text{jt}}(A, A) + \pi_{\text{jt}}^{\text{dep}}(B, B) * Q_{\text{jt}}(B, B) \tag{8}$$

$$= \pi_{\text{jt}}^{\text{ind}}(A, A) * Q_{\text{jt}}(A, A) = \mathbb{E}_{\pi_{\text{jt}}^{\text{ind}}}[Q_{\text{jt}}]. \tag{9}$$

Formally, we have the following theorem.

**Theorem 2.** *For any dependent joint policy $\pi_{\text{jt}}^{\text{dep}}$ that involves dependency among agents, there exists an independent joint policy $\pi_{\text{jt}}^{\text{ind}}$ that does not involve dependency among agents, such that $V_{\pi_{\text{jt}}^{\text{dep}}}(s) = V_{\pi_{\text{jt}}^{\text{ind}}}(s)$ for any state $s \in S$.*

*Proof.* See Appendix B. $\qquad\square$

***Note that the independent counterpart of the optimal dependent joint policy may not be directly learned by FOP, as shown in Figure 1. Therefore, we need to explicitly learn the optimal dependent joint policy to obtain its independent counterpart.***

### 3.3 MACPF FRAMEWORK

With Theorem 1 and 2, we are ready to present the learning framework of MACPF, as illustrated in Figure 3, for simultaneously learning the dependent joint policy and its independent counterpart.

In MACPF, each agent $i$ has an independent local policy $\pi_i^{\text{ind}}(a_i \,|\, \text{s}; \theta_i)$ parameterized by $\theta_i$ and a dependency policy correction $b_i^{\text{dep}}(a_i \,|\, \text{s}, a_{<i}; \phi_i)$ parameterized by $\phi_i$, which together constitute a dependent local policy $\pi_i^{\text{dep}}(a_i \,|\, \text{s}, a_{<i})$[1]. So, we have:

$$\pi_i^{\text{dep}}(a_i \,|\, \text{s}, a_{<i}) = \pi_i^{\text{ind}}(a_i \,|\, \text{s}; \theta_i) + b_i^{\text{dep}}(a_i \,|\, \text{s}, a_{<i}; \phi_i) \tag{10}$$

$$\pi_{\text{jt}}^{\text{dep}}(\text{s}, \boldsymbol{a}) = \prod_{i=1}^{N} \pi_i^{\text{dep}}(a_i \,|\, \text{s}, a_{<i}) \tag{11}$$

$$\pi_{\text{jt}}^{\text{ind}}(\text{s}, \boldsymbol{a}) = \prod_{i=1}^{N} \pi_i^{\text{ind}}(a_i \,|\, \text{s}; \theta_i). \tag{12}$$

Similarly, each agent $i$ also has an independent local critic $Q_i^{\text{ind}}(a_i \,|\, \text{s}; \psi_i)$ parameterized by $\psi_i$ and a dependency critic correction $c_i^{\text{dep}}(a_i \,|\, \text{s}, a_{<i}; \omega_i)$ parameterized by $\omega_i$, which together constitute a dependent local critic $Q_i^{\text{dep}}(a_i \,|\, \text{s}, a_{<i})$. Given all $Q_i^{\text{ind}}$ and $Q_i^{\text{dep}}$, we use a mixer network, $\text{Mixer}(\cdot; \Theta)$ parameterized by $\Theta$, to get $Q_{\text{jt}}^{\text{dep}}$ and $Q_{\text{jt}}^{\text{ind}}$ as follows,

$$Q_i^{\text{dep}}(a_i \,|\, \text{s}, a_{<i}) = Q_i^{\text{ind}}(a_i \,|\, \text{s}; \psi_i) + c_i^{\text{dep}}(a_i \,|\, \text{s}, a_{<i}; \omega_i) \tag{13}$$

$$Q_{\text{jt}}^{\text{dep}}(\text{s}, \boldsymbol{a}) = \text{Mixer}([Q_i^{\text{dep}}(a_i \,|\, \text{s}, a_{<i})]_{i=1}^{N}, \text{s}; \Theta) \tag{14}$$

$$Q_{\text{jt}}^{\text{ind}}(\text{s}, \boldsymbol{a}) = \text{Mixer}([Q_i^{\text{ind}}(a_i \,|\, \text{s}; \psi_i)]_{i=1}^{N}, \text{s}; \Theta). \tag{15}$$

$Q_i^{\text{dep}}$, $Q_i^{\text{ind}}$, and $\text{Mixer}$ are learned by minimizing the TD error,

$$\mathcal{L}^{\text{dep}}([\omega_i]_{i=1}^{N}, \Theta) = \mathbb{E}_{\mathcal{D}} \left[ \left( Q_{\text{jt}}^{\text{dep}}(\text{s}, \boldsymbol{a}) - \left( r + \gamma \big( \hat{Q}_{\text{jt}}^{\text{dep}}(\text{s}', \boldsymbol{a}') - \alpha \log \pi_{\text{jt}}^{\text{dep}}(\boldsymbol{a}' \,|\, \text{s}') \big) \right) \right)^2 \right] \tag{16}$$

$$\mathcal{L}^{\text{ind}}([\psi_i]_{i=1}^{N}, \Theta) = \mathbb{E}_{\mathcal{D}} \left[ \left( Q_{\text{jt}}^{\text{ind}}(\text{s}, \boldsymbol{a}) - \left( r + \gamma \big( \hat{Q}_{\text{jt}}^{\text{ind}}(\text{s}', \boldsymbol{a}') - \alpha \log \pi_{\text{jt}}^{\text{ind}}(\boldsymbol{a}' \,|\, \text{s}') \big) \right) \right)^2 \right], \tag{17}$$

---

[1]The logit of $\pi_i^{\text{ind}}(a_i \,|\, \text{s}; \theta_i)$ is first added with $b_i^{\text{dep}}(a_i \,|\, \text{s}, a_{<i}; \phi_i)$ to get the logit of $\pi_i^{\text{dep}}(a_i \,|\, \text{s}, a_{<i})$, then softmax is used over this combined logit to get $\pi_i^{\text{dep}}(a_i \,|\, \text{s}, a_{<i})$.

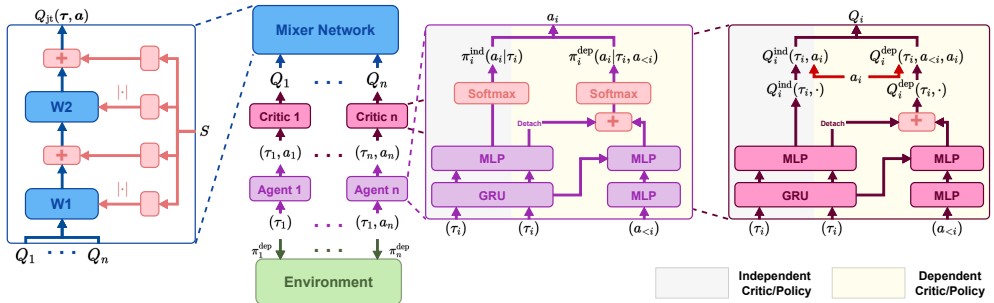

Figure 3: Learning framework of MACPF, where each agent $i$ has four modules: an independent local policy $\pi_i^{\text{ind}}(\cdot; \theta_i)$, a dependency policy correction $b_i^{\text{dep}}(\cdot; \phi_i)$, an independent local critic $Q_i^{\text{ind}}(\cdot; \psi_i)$, and a dependency critic correction $c_i^{\text{dep}}(\cdot, \omega_i)$.

where $\mathcal{D}$ is the replay buffer collected by $\pi_{\text{jt}}^{\text{dep}}$, $\hat{Q}$ is the target network, and $\boldsymbol{a}'$ is sampled from the current $\pi_{\text{jt}}^{\text{dep}}$ and $\pi_{\text{jt}}^{\text{ind}}$, respectively. To ensure the independent joint policy $\pi_{\text{jt}}^{\text{ind}}$ has the same performance as $\pi_{\text{jt}}^{\text{dep}}$, the *same batch* sampled from $\mathcal{D}$ is used to compute both $\mathcal{L}^{\text{dep}}$ and $\mathcal{L}^{\text{ind}}$. It is worth noting that the gradient of $\mathcal{L}^{\text{dep}}$ only updates $[c_i^{\text{dep}}]_{i=1}^N$, while the gradient of $\mathcal{L}^{\text{ind}}$ only updates $[Q_i^{\text{ind}}]_{i=1}^N$. Then, $\pi_i^{\text{dep}}$ and $\pi_i^{\text{ind}}$ are updated by minimizing KL-divergence as follows,

$$\mathcal{J}^{\text{dep}}(\phi_i) = \mathbb{E}_{\mathcal{D}, a_{<i} \sim \pi_{<i}^{\text{dep}}, a_i \sim \pi_i^{\text{dep}}}[\alpha_i \log \pi_i^{\text{dep}}(a_i \,|\, \text{s}, a_{<i}) - Q_i^{\text{dep}}(a_i \,|\, \text{s}, a_{<i})] \qquad (18)$$

$$\mathcal{J}^{\text{ind}}(\theta_i) = \mathbb{E}_{\mathcal{D}, a_i \sim \pi_i^{\text{ind}}}[\alpha_i \log \pi_i^{\text{ind}}(a_i \,|\, \text{s}; \theta_i) - Q_i^{\text{ind}}(a_i \,|\, \text{s}; \psi_i)]. \qquad (19)$$

Similarly, the gradient of $\mathcal{J}^{\text{dep}}$ only updates $b_i^{\text{dep}}$ and the gradient of $\mathcal{J}^{\text{ind}}$ only updates $\pi_i^{\text{ind}}$. For computing $\mathcal{J}^{\text{dep}}$, $a_{<i}$ is sampled from their current policies $\pi_{<i}^{\text{dep}}$.

The purpose of learning $\pi_{\text{jt}}^{\text{ind}}$ is to enable decentralized execution while achieving the same performance as $\pi_{\text{jt}}^{\text{dep}}$. Therefore, a certain level of coupling has to be assured between $\pi_{\text{jt}}^{\text{ind}}$ and $\pi_{\text{jt}}^{\text{dep}}$. First, motivated by Figure 2, we constitute the dependent policy as a combination of an independent policy and a dependency policy correction, similarly for the local critic. Second, as aforementioned, the replay buffer $\mathcal{D}$ is collected by $\pi_{\text{jt}}^{\text{dep}}$, which implies $\pi_{\text{jt}}^{\text{dep}}$ is the behavior policy and the learning of $\pi_{\text{jt}}^{\text{ind}}$ is offline. Third, we use the same Mixer to compute $Q_{\text{jt}}^{\text{dep}}$ and $Q_{\text{jt}}^{\text{ind}}$. The performance comparison between $\pi_{\text{jt}}^{\text{dep}}$ and $\pi_{\text{jt}}^{\text{ind}}$ will be studied by experiments.

## 4 RELATED WORK

**Multi-agent policy gradient.** In multi-agent policy gradient, a centralized value function is usually learned to evaluate current joint policy and guide the update of each local policy. Most multi-agent policy gradient methods can be considered as an extension of policy gradient from RL to MARL. For example, MAPPDG (Lowe et al., 2017) extends DDPG (Lillicrap et al., 2015), PS-TRPO(Gupta et al., 2017) and MATRPO (Kuba et al., 2021) extend TRPO (Schulman et al., 2015), and MAPPO (Yu et al., 2021) extends PPO (Schulman et al., 2017). Some methods additionally address multi-agent credit assignment by policy gradient, *e.g.*, counterfactual policy gradient (Foerster et al., 2018) or difference rewards policy gradient (Castellini et al., 2021; Li et al., 2022).

**Value decomposition.** Instead of providing gradients for local policies, in value decomposition, the centralized value function, usually a joint Q-function, is directly decomposed into local utility functions. Many methods have been proposed as different interpretations of Individual-Global-Maximum (IGM), which indicates the consistency between optimal local actions and optimal joint action. VDN (Sunehag et al., 2018) and QMIX (Rashid et al., 2018) give sufficient conditions for IGM by additivity and monotonicity, respectively. QTRAN (Son et al., 2019) transforms IGM into optimization constraints, while QPLEX (Wang et al., 2020a) takes advantage of duplex dueling architecture to guarantee IGM. Recent studies (Su et al., 2021; Wang et al., 2020b; Zhang et al., 2021; Su & Lu, 2022) combine value decomposition with policy gradient to learn stochastic policies, which

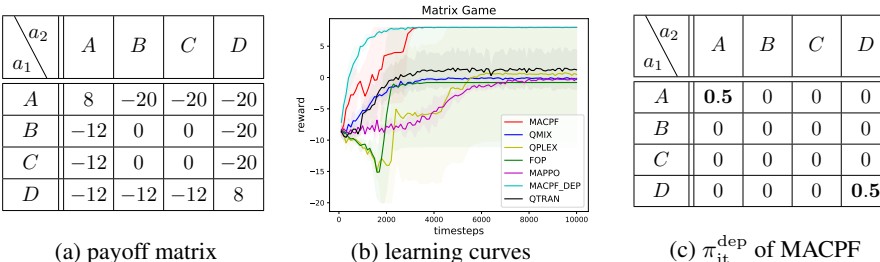

(a) payoff matrix · (b) learning curves · (c) $\pi_{\mathrm{jt}}^{\mathrm{dep}}$ of MACPF

Figure 4: A matrix game that has two optimal joint actions: (a) payoff matrix; (b) learning curves of different methods; (c) the learned dependent joint policy of MACPF.

are more desirable in partially observable environments. However, most research in this category does not guarantee optimality, while our method enables agents to learn the optimal joint policy.

**Coordination graph.** In coordination graph (Guestrin et al., 2002) methods (Böhmer et al., 2020; Wang et al., 2021b; Yang et al., 2022), the interactions between agents are considered as part of value decomposition. Specifically, the joint Q-function is decomposed into the combination of utility functions and payoff functions. The introduction of payoff functions increases the expressiveness of the joint Q-function and considers at least pair-wise dependency among agents, which is similar to our algorithm, where the complete dependency is considered. However, to get the joint action with the maximum Q-value, communication between agents is required in execution in coordination graph methods, while our method still fulfills fully decentralized execution.

**Coordinated exploration.** One of the benefits of considering dependency is coordinated exploration. From this perspective, our method might be seen as a relative of coordinated exploration methods (Mahajan et al., 2019; Iqbal & Sha, 2019; Zheng et al., 2021). In MAVEN (Mahajan et al., 2019), a shared latent variable is used to promote committed, temporally extended exploration. In EMC (Zheng et al., 2021), the intrinsic reward based on the prediction error of individual Q-values is used to induce coordinated exploration. It is worth noting that our method does not conflict with coordinated exploration methods and can be used simultaneously as our method is a base cooperative MARL algorithm. However, such a combination is beyond the scope of this paper.

## 5 EXPERIMENTS

In this section, we evaluate MACPF in three different scenarios. One is a simple yet challenging matrix game, which we use to verify whether MACPF can indeed converge to the optimal joint policy. Then, we evaluate MACPF on two popular cooperative MARL scenarios: StarCraft Multi-Agent Challenge (SMAC) (Samvelyan et al., 2019) and MPE (Lowe et al., 2017), comparing it against QMIX (Rashid et al., 2018), QPLEX (Wang et al., 2020a), FOP (Zhang et al., 2021), and MAPPO (Yu et al., 2021). More details about experiments and hyperparameters are included in Appendix C. All results are presented using the mean and standard deviation of five runs with different random seeds. In SMAC experiments, for visual clarity, we plot the curves with the moving average of a window size of five and half standard deviation.

### 5.1 MATRIX GAME

In this matrix game, we have two agents. Each can pick one of the four actions and get a reward based on the payoff matrix depicted in Figure 4a. Unlike the non-monotonic matrix game in QTRAN (Son et al., 2019), where there is only one optimal joint action, we have two optimal joint actions in this game, making this scenario much more challenging for many cooperative MARL algorithms.

As shown in Figure 4b, general value decomposition methods, QMIX, QPLEX, and FOP, fail to learn the optimal coordinated strategy in most cases. The same negative result can also be observed for MAPPO. For general MARL algorithms, since agents are fully independent of each other when making decisions, they may fail to converge to the optimal joint action, which eventually leads to a suboptimal joint policy. As shown in Figure 4b, QMIX and MAPPO fail to converge to the optimal policy but find a suboptimal policy in all the seeds, while QPLEX, QTRAN, and FOP find the optima by chance (*i.e.*, 60% for QPLEX, 20% for QTRAN, and 40% for FOP). This is because, in QMIX, the

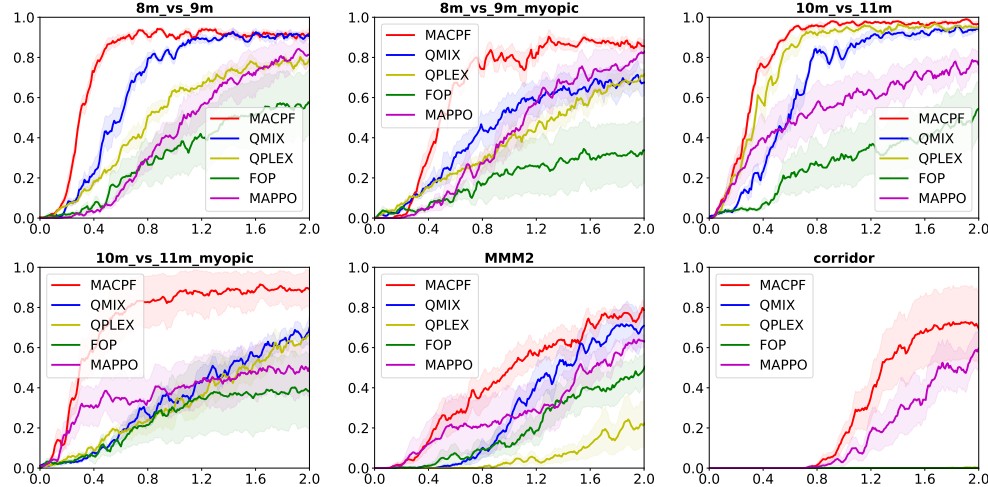

Figure 5: Learning curves of all the methods in six maps of SMAC, where the unit of x-axis is 1M timesteps and y-axis represents the win rate of each map.

mixer network is purely a function of state and the input utility functions that are fully independent of each other. Thus it considers no dependency at all and cannot solve this game where dependency has to be considered. For QPLEX and FOP, since the joint action is considered as the input of their mixer network, the dependency among agents may be implicitly considered, which leads to the case where they can find the optima by chance. However, since the dependency is not considered explicitly, there is also a possibility that the mixer network misinterprets the dependency, which makes QPLEX and FOP sometimes find even worse policies than QMIX (20% for QPLEX and 40% for FOP). For QTRAN, it always finds at least the suboptimal policy in all the seeds. However, its optimality largely relies on the learning of its $V_{jt}$, which is very unstable, so it also only finds the optima by chance.

For the dependent joint policy $\pi_{jt}^{dep}$ of MACPF, the local policy of the second agent depends on the action of the first agent. As a result, we can see from Figure 4b that $\pi_{jt}^{dep}$ (denoted as MACPF_DEP) always converges to the highest return. We also notice that in Figure 4c, $\pi_{jt}^{dep}$ indeed captures two optimal joint actions. Unlike QMIX, QPLEX, and FOP, the mixer network in MACPF is a function of state and the input utility functions $Q_i^{dep}(a_i | s, a_{<i})$ that are properly dependent on each other, so the dependency among agents is explicitly considered. More importantly, the learned independent joint policy $\pi_{jt}^{ind}$ of MACPF, denoted as MACPF in Figure 4b, always converges to the optimal joint policy. *Note that in the rest of this section, the performance of MACPF is achieved by the learned $\pi_{jt}^{ind}$, unless stated otherwise.*

## 5.2 SMAC

Further, we evaluate MACPF on SMAC. Maps used in our experiment include two hard maps (8m_vs_9m, 10m_vs_11m), and two super-hard maps (MMM2, corridor). We also consider two challenging customized maps (8m_vs_9m_myopic, 10m_vs_11m_myopic), where the sight range of each agent is reduced from 9 to 6, and the information of allies is removed from the observation of agents. These changes are adopted to increase the difficulty of coordination in the original maps. Results are shown in Figure 5. In general, MACPF outperforms the baselines in all six maps. In hard maps, MACPF outperforms the baselines mostly in convergence speed, while in super-hard maps, MACPF outperforms other algorithms in either convergence speed or performance. Especially in corridor, when other value decomposition algorithms fail to learn any meaningful joint policies, MACPF obtains a winning rate of almost 70%. In the two more challenging maps, the margin between MACPF and the baselines becomes much larger than that in the original maps. These results show that MACPF can better handle complex cooperative tasks and learn coordinated strategies by introducing dependency among agents even when the task requires stronger coordination.

We compare MACFP with the baselines in 18 maps totally. Their final performance is summarized in Appendix D. *The win rate of MACFP is higher than or equivalent to the best baseline in 16 out of 18 maps, while QMIX, QPLEX, MAPPO, and FOP are respectively 7/18, 8/18, 9/18, and 5/18.*

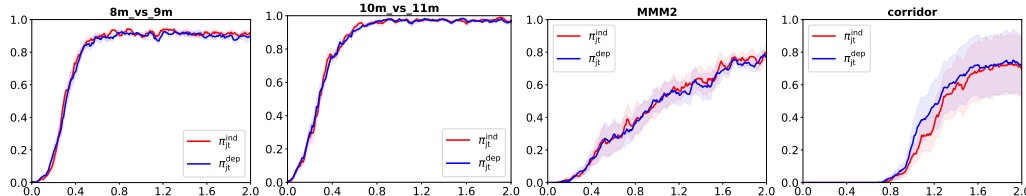

Figure 6: Performance of $\pi_{\text{jt}}^{\text{dep}}$ and $\pi_{\text{jt}}^{\text{ind}}$ during training in four maps of SMAC, where the unit of x-axis is 1M timesteps and y-axis represents the win rate of each map.

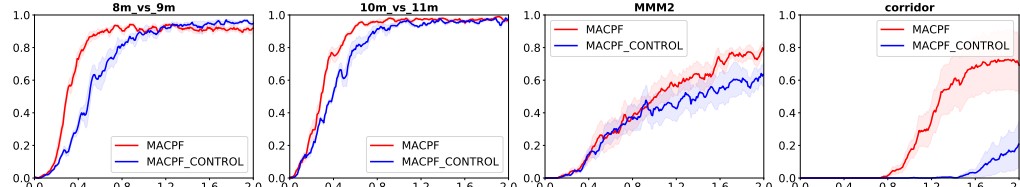

Figure 7: Ablation study in four maps of SMAC, where the unit of x-axis is 1M timesteps and y-axis represents the win rate of each map.

**Dependent and Independent Joint Policy.** As discussed in Section 3.2, the learned independent joint policy of MACPF should not only enable decentralized execution but also match the performance of dependent joint policy, as verified in the matrix game. *What about complex environments like SMAC?* As shown in Figure 6, we track the evaluation result of both $\pi_{\text{jt}}^{\text{ind}}$ and $\pi_{\text{jt}}^{\text{dep}}$ during training. As we can see, their performance stays at the same level throughout training.

**Ablation Study.** Without learning a dependent joint policy to interact with the environment, our algorithm degenerates to FOP. However, since our factorization of $Q_{\text{jt}}$ is induced from the chain rule factorization of joint probability (5), we use a mixer network different from FOP (the reason is discussed and verified in Appendix E). Here we present an ablation study to further show that the improvement of MACPF is indeed induced by introducing the dependency among agents. In Figure 7, MACPF_CONTROL represents an algorithm where all other perspectives are the same as MACPF, except no dependent joint policy is learned. As shown in Figure 7, MACPF outperforms MACPF_CONTROL in all four maps, demonstrating that the performance improvement is indeed achieved by introducing the dependency among agents.

### 5.3 MPE

We further evaluate MACPF on three MPE tasks, including simple spread, formation control, and line control (Agarwal et al., 2020). As shown in Table 1, MACPF outperforms the baselines in all three tasks. A large margin can be observed in simple spread, while only a minor difference can be observed in the other two. This result may indicate that these MPE tasks are not challenging enough for strong MARL algorithms.

Table 1: Average rewards per episode on three MPE tasks.

| Algorithms / Tasks | MACPF | QMIX | QPLEX | FOP | MAPPO |
|---|---|---|---|---|---|
| Simple Spread | **-118.24**±2.74 | -145.93±21.09 | -122.50±2.58 | -125.19±5.42 | -166.75±23.44 |
| Formation Control | **-15.79**±0.16 | -16.11±0.30 | -16.10±0.28 | −15.84±0.19 | -21.71±1.69 |
| Line Control | **-19.60**±0.33 | -20.12±0.21 | -20.17±0.26 | -19.78±0.27 | -24.47±2.54 |

### 6 CONCLUSION

We have proposed MACPF, where dependency among agents is introduced to enable more centralized training. By conditional factorized soft policy iteration, we show that dependent local policies provably converge to the optimum. To fulfill decentralized execution, we represent dependent local policies as a combination of independent local policies and dependency policy corrections, such that independent local policies can achieve the same level of expected return as dependent ones. Empirically, we show that MACPF can obtain the optimal joint policy in a simple yet challenging matrix game while baselines fail and MACPF also outperforms the baselines in SMAC and MPE.

ACKNOWLEDGMENTS

This work was supported in part by NSFC (under grant 62250068) and Tencent.

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

## A  PROOF OF THEOREM 1

In this subsection, we incorporate dependency among agents into the standard soft policy iteration and prove that this modified soft policy iteration converges to the optimal joint policy.

For *soft policy evaluation*, we will repeatedly apply soft Bellman operator $\Gamma_{\pi_{\mathrm{jt}}}$ to $Q_{\mathrm{jt}}^{\pi_{\mathrm{jt}}}$ until convergence, where:

$$\Gamma_{\pi_{\mathrm{jt}}} Q_{\mathrm{jt}}(\mathrm{s}_t, \boldsymbol{a}_t) := r_t + \gamma \, \mathbb{E}_{\mathrm{s}_{t+1}}[V_{\mathrm{jt}}(\mathrm{s}_{t+1})] \tag{20}$$

$$V_{\mathrm{jt}}(\mathrm{s}_t) = \mathbb{E}_{\pi_{\mathrm{jt}}}[Q_{\mathrm{jt}}(\mathrm{s}_t, \boldsymbol{a}_t) - \alpha \log \pi_{\mathrm{jt}}(\boldsymbol{a}_t | \mathrm{s}_t)]. \tag{21}$$

In this way, as shown in Lemma A.1, we can get $Q_{\mathrm{jt}}^{\pi_{\mathrm{jt}}}$ for any joint policy $\pi_{\mathrm{jt}}$.

**Lemma A.1** (**Joint Soft Policy Evaluation**). *Consider the modified soft Bellman backup operator $\Gamma_{\pi_{\mathrm{jt}}}$ and a mapping $Q_{\mathrm{jt}}^0 : \mathrm{S} \times A \to \mathbb{R}$ with $|A| < \infty$, and define $Q_{\mathrm{jt}}^{k+1} = \Gamma_{\pi_{\mathrm{jt}}} Q_{\mathrm{jt}}^k$. Then, the sequence $Q_{\mathrm{jt}}^k$ will converge to the joint soft Q-function of $\pi_{\mathrm{jt}}$ as $k \to \infty$.*

*Proof.* First, define the entropy augmented reward as:

$$r_{\pi_{\mathrm{jt}}}(\mathrm{s}_t, \boldsymbol{a}_t) := r(\mathrm{s}_t, \boldsymbol{a}_t) + \mathbb{E}_{\mathrm{s}_{t+1}}[\mathcal{H}(\pi_{\mathrm{jt}}(\cdot | \mathrm{s}_{t+1}))].$$

Then, rewrite the update rule as:

$$Q_{\mathrm{jt}}(\mathrm{s}_t, \boldsymbol{a}_t) \leftarrow r_{\pi_{\mathrm{jt}}}(\mathrm{s}_t, \boldsymbol{a}_t) + \gamma \, \mathbb{E}_{\mathrm{s}_{t+1}, \boldsymbol{a}_{t+1} \sim \pi_{\mathrm{jt}}}[Q_{\mathrm{jt}}(\mathrm{s}_{t+1}, \boldsymbol{a}_{t+1})].$$

Last, apply the standard convergence results for policy evaluation (Sutton & Barto, 2018). $\square$

After we get $Q_{\mathrm{jt}}^{\pi_{\mathrm{jt}}}$, we will make a one-step improvement for the joint policy. First, we restrict the local policy $\pi_i$ of each agent $i$ to some set of policies $\Pi_i$ and update the local policy according to the following optimization problem:

$$\pi_i^{\mathrm{new}} = \underset{\pi_i' \in \Pi_i}{\arg\min} \underbrace{\mathbb{E}_{a_{<i} \sim \pi_{<i}^{\mathrm{new}}} \left[ D_{\mathrm{KL}} \left( \pi_i'(a_i | \mathrm{s}, a_{<i}) \, \| \, \exp \left( \frac{1}{\alpha_i} \big( Q_i^{\pi_i^{\mathrm{old}}}(\mathrm{s}, a_{<i}, a_i) - V_i^{\pi_i^{\mathrm{old}}}(\mathrm{s}, a_{<i}) \big) \right) \right) \right]}_{J_{\pi_i^{\mathrm{old}}, a_{<i}}(\pi_i'(a_i | \mathrm{s}, a_{<i}))}. \tag{22}$$

Based on *individual conditional soft policy improvement*, we will show that the newly projected joint soft policy has a higher state-action value than the old joint soft policy with respect to the maximum-entropy RL objective.

**Lemma A.2** (**Individual Conditional Soft Policy Improvement**). *Let $\pi_i^{\mathrm{old}} \in \Pi_i$ and $\pi_i^{\mathrm{new}}$ be the optimizer of the minimization problem in (22). Then, we have $Q_{\mathrm{jt}}^{\pi_{\mathrm{jt}}^{\mathrm{new}}}(\mathrm{s}_t, \boldsymbol{a}_t) \geq Q_{\mathrm{jt}}^{\pi_{\mathrm{jt}}^{\mathrm{old}}}(\mathrm{s}_t, \boldsymbol{a}_t)$ for all $(\mathrm{s}_t, \boldsymbol{a}_t) \in \mathrm{S} \times A$ with $|A| < \infty$, where $\pi_{\mathrm{jt}}^{\mathrm{old}}(\boldsymbol{a} | \mathrm{s}) = \prod_{i=1}^N \pi_i^{\mathrm{old}}(a_i | \mathrm{s}, a_{<i})$ and $\pi_{\mathrm{jt}}^{\mathrm{new}}(\boldsymbol{a} | \mathrm{s}) = \prod_{i=1}^N \pi_i^{\mathrm{new}}(a_i | \mathrm{s}, a_{<i})$.*

*Proof.* Let $Q_i^{\pi_i^{\mathrm{old}}}$ and $V_i^{\pi_i^{\mathrm{old}}}$ be the corresponding soft state-action value and soft state value of individual policy $\pi_i^{\mathrm{old}}$. First, considering that $J_{\pi_i^{\mathrm{old}}, a_{<i}}(\pi_i^{\mathrm{new}}(a_i | \mathrm{s}, a_{<i})) \leq J_{\pi_i^{\mathrm{old}}, a_{<i}}(\pi_i^{\mathrm{old}}(a_i | \mathrm{s}, a_{<i}))$. Then, we have:

$$\begin{aligned} &\mathbb{E}_{a_i \sim \pi_i^{\mathrm{new}}, a_{<i} \sim \pi_{<i}^{\mathrm{new}}}[\alpha_i \log \pi_i^{\mathrm{new}}(a_i | \mathrm{s}, a_{<i}) - Q_i^{\pi_i^{\mathrm{old}}}(\mathrm{s}, a_{<i}, a_i) + V_i^{\pi_i^{\mathrm{old}}}(\mathrm{s}, a_{<i})] \\ &\leq \mathbb{E}_{a_i \sim \pi_i^{\mathrm{old}}, a_{<i} \sim \pi_{<i}^{\mathrm{new}}}[\alpha_i \log \pi_i^{\mathrm{old}}(a_i | \mathrm{s}, a_{<i}) - Q_i^{\pi_i^{\mathrm{old}}}(\mathrm{s}, a_{<i}, a_i) + V_i^{\pi_i^{\mathrm{old}}}(\mathrm{s}, a_{<i})]. \end{aligned} \tag{23}$$

Since $V_i^{\pi_i^{\mathrm{old}}}$ depends only on s and $a_{<i}$, where:

$$\mathbb{E}_{a_{<i} \sim \pi_{<i}^{\mathrm{new}}}[V_i^{\pi_i^{\mathrm{old}}}(\mathrm{s}, a_{<i})] = \mathbb{E}_{a_{<i} \sim \pi_{<i}^{\mathrm{new}}, a_i \sim \pi_i^{\mathrm{old}}}[Q_i^{\pi_i^{\mathrm{old}}}(\mathrm{s}, a_{<i}, a_i) - \alpha_i \log \pi_i^{\mathrm{old}}(a_i | \mathrm{s}, a_{<i})]. \tag{24}$$

By deducing (24) from both sides of (23), we have:

$$\mathbb{E}_{a_i \sim \pi_i^{\mathrm{new}}, a_{<i} \sim \pi_{<i}^{\mathrm{new}}}[Q_i^{\pi_i^{\mathrm{old}}}(\mathrm{s}, a_{<i}, a_i) - \alpha_i \log \pi_i^{\mathrm{new}}(a_i | \mathrm{s}, a_{<i})] \geq \mathbb{E}_{a_{<i} \sim \pi_{<i}^{\mathrm{new}}}[V_i^{\pi_i^{\mathrm{old}}}(\mathrm{s}, a_{<i})]. \tag{25}$$

And since

$$\pi_{\text{jt}}^{\text{new}} = \exp\left(\frac{1}{\alpha}\left(Q_{\text{jt}}^{\pi_{\text{jt}}^{\text{old}}}(s, \boldsymbol{a}) - V_{\text{jt}}^{\pi_{\text{jt}}^{\text{old}}}(s)\right)\right)$$

$$\pi_i^{\text{new}} = \exp\left(\frac{1}{\alpha}\left(Q_i^{\pi_i^{\text{old}}}(s, a_{<i}, a_i) - V_i^{\pi_i^{\text{old}}}(s, a_{<i})\right)\right)$$

$$\pi_{\text{jt}}^{\text{new}}(\boldsymbol{a}|s) = \prod_{i=1}^{N} \pi_i^{\text{new}}(a_i|s, a_{<i}),$$

we can have:

$$Q_{\text{jt}}^{\pi_{\text{jt}}^{\text{old}}}(s, \boldsymbol{a}) = \sum_{i=1}^{N} \frac{\alpha}{\alpha_i}[Q_i^{\pi_i^{\text{old}}}(s, a_{<i}, a_i) - V_i^{\pi_i^{\text{old}}}(s, a_{<i})] + V_{\text{jt}}^{\pi_{\text{jt}}^{\text{old}}}(s).$$

Then we have

$$\mathbb{E}_{\boldsymbol{a}\sim\pi_{\text{jt}}^{\text{new}}}[Q_{\text{jt}}^{\pi_{\text{jt}}^{\text{old}}}(s, \boldsymbol{a}) - \alpha\log\pi_{\text{jt}}^{\text{new}}(\boldsymbol{a}|s)]$$

$$= \mathbb{E}_{\boldsymbol{a}\sim\pi_{\text{jt}}^{\text{new}}}\left[\sum_{i=1}^{N}\frac{\alpha}{\alpha_i}[Q_i^{\pi_i^{\text{old}}}(s, a_{<i}, a_i) - V_i^{\pi_i^{\text{old}}}(s, a_{<i})] + V_{\text{jt}}^{\pi_{\text{jt}}^{\text{old}}}(s) - \alpha\log\pi_{\text{jt}}^{\text{new}}(\boldsymbol{a}|s)\right]$$

$$= \sum_{i=1}^{N}\mathbb{E}_{\boldsymbol{a}\sim\pi_{\text{jt}}^{\text{new}}, a_{<i}\sim\pi_{<i}^{\text{new}}}\left[\frac{\alpha}{\alpha_i}[Q_i^{\pi_i^{\text{old}}}(s, a_{<i}, a_i) - V_i^{\pi_i^{\text{old}}}(s, a_{<i}) - \alpha_i\log\pi_i^{\text{new}}(a_i|s, a_{<i})]\right] + V_{\text{jt}}^{\pi_{\text{jt}}^{\text{old}}}(s)$$

$$\geq V_{\text{jt}}^{\pi_{\text{jt}}^{\text{old}}}(s), \tag{26}$$

where the inequality is from plugging in (25).

Last, considering the soft bellman equation, the following holds:

$$Q_{\text{jt}}^{\pi_{\text{jt}}^{\text{old}}}(s_t, \boldsymbol{a}_t) = r_t + \gamma\,\mathbb{E}_{s_{t+1}}[V_{\text{jt}}^{\pi_{\text{jt}}^{\text{old}}}(s)]$$

$$\leq r_t + \gamma\,\mathbb{E}_{s_{t+1}}[\mathbb{E}_{\boldsymbol{a}\sim\pi_{\text{jt}}^{\text{new}}}[Q_{\text{jt}}^{\pi_{\text{jt}}^{\text{old}}}(s_{t+1}, \boldsymbol{a}_{t+1}) - \alpha\log\pi_{\text{jt}}^{\text{new}}(\boldsymbol{a}_{t+1}|s_{t+1})]]$$

$$\vdots$$

$$\leq Q_{\text{jt}}^{\pi_{\text{jt}}^{\text{new}}}(s_t, \boldsymbol{a}_t),$$

where we have repeatedly expanded $Q_{\text{jt}}^{\pi_{\text{jt}}^{\text{old}}}$ on the RHS by applying the soft Bellman equation and the bound in (26). $\square$

*Conditional factorized soft policy iteration* alternates between joint soft policy evaluation and individual conditional soft policy improvement, and provably converges to the global optimum, as shown in Theorem 1.

**Theorem 1** (**Conditional Factorized Soft Policy Iteration**). *For any joint policy $\pi_{\text{jt}}$, if we repeatedly apply joint soft policy evaluation and individual conditional soft policy improvement from $\pi_i \in \Pi_i$. Then the joint policy $\pi_{\text{jt}}(\boldsymbol{a}|s) = \prod_{i=1}^{n}\pi_i(a_i|s, a_{<i})$ will eventually converge to $\pi_{\text{jt}}^*$, such that $Q_{\text{jt}}^{\pi_{\text{jt}}^*}(s, \boldsymbol{a}) \geq Q_{\text{jt}}^{\pi_{\text{jt}}}(s, \boldsymbol{a})$ for all $\pi_{\text{jt}}$, assuming $|A| < \infty$.*

*Proof.* First, by Lemma A.2, the sequence $\{\pi_{\text{jt}}^k\}$ monotonically improves with $Q_{\text{jt}}^{\pi_{\text{jt}}^{k+1}} \geq Q_{\text{jt}}^{\pi_{\text{jt}}^k}$. Since both the reward and entropy are bounded, then $Q_{\text{jt}}^{\pi_{\text{jt}}^k}$ is bounded. Thus, this sequence must converge to some $\pi_{\text{jt}}^*$. Then, at convergence, we have the following inequality:

$$J_{\pi_{\text{jt}}^*}(\pi_{\text{jt}}^*(\cdot|s)) \leq J_{\pi_{\text{jt}}^*}(\pi_{\text{jt}}(\cdot|s)), \forall \pi_{\text{jt}} \neq \pi_{\text{jt}}^*.$$

Using the same iterative argument as in the proof of Lemma A.2, we get $Q_{\text{jt}}^{\pi_{\text{jt}}^*}(s, \boldsymbol{a}) \geq Q_{\text{jt}}^{\pi_{\text{jt}}}(s, \boldsymbol{a})$ for all $(s, \boldsymbol{a}) \in S \times A$. That is, the soft value of any other policy $\pi_{\text{jt}}$ is lower than that of the converged policy $\pi_{\text{jt}}^*$. Therefore, $\pi_{\text{jt}}^*$ is optimal in $\Pi_1 \times \cdots \times \Pi_N$. $\square$

## B  PROOF OF THEOREM 2

**Theorem 2.** *For any dependent joint policy $\pi_{\mathrm{jt}}^{\mathrm{dep}}$ that involves dependency among agents, there exists an independent joint policy $\pi_{\mathrm{jt}}^{\mathrm{ind}}$ that does not involve dependency among agents, such that $V_{\pi_{\mathrm{jt}}^{\mathrm{dep}}}(\mathrm{s}) = V_{\pi_{\mathrm{jt}}^{\mathrm{ind}}}(\mathrm{s})$ for any state $\mathrm{s} \in \mathrm{S}$.*

*Proof.* For a dependent joint policy $\pi_{\mathrm{jt}}^{\mathrm{dep}}$ that involves dependency among agents, let $\max_{\boldsymbol{a}} Q_{\pi_{\mathrm{jt}}^{\mathrm{dep}}} = A$ and $\min_{\boldsymbol{a}} Q_{\pi_{\mathrm{jt}}^{\mathrm{dep}}} = B$, we have $A \leq V_{\pi_{\mathrm{jt}}^{\mathrm{dep}}}(\mathrm{s}) \leq B$. Then, we can construct the following independent joint policy $\pi_{\mathrm{jt}}^{\mathrm{ind}}$:

$$\pi_{\mathrm{jt}}^{\mathrm{ind}} = \prod_{i=1}^{N} \pi_i = \prod_{i=1}^{N} \mathbf{1}[a_i = \arg\max Q_{\pi_{\mathrm{jt}}^{\mathrm{dep}}}[i]].$$

For such an independent joint policy $\pi_{\mathrm{jt}}^{\mathrm{ind}}$, we have $\sum_{\boldsymbol{a}} \pi_{\mathrm{jt}}^{\mathrm{ind}} Q_{\pi_{\mathrm{jt}}^{\mathrm{dep}}} = A$. Similarly, we can also construct another independent joint policy, such that $\sum_{\boldsymbol{a}} \pi_{\mathrm{jt}}^{\mathrm{ind}} Q_{\pi_{\mathrm{jt}}^{\mathrm{dep}}} = B$. Based on the generalized intermediate value theorem (Munkres, 2000), We can have that for any dependent joint policy $\pi_{\mathrm{jt}}^{\mathrm{dep}}$, there exist an independent joint policy $\pi_{\mathrm{jt}}^{\mathrm{ind}}$ such that:

$$V_{\pi_{\mathrm{jt}}^{\mathrm{dep}}} = \sum_{\boldsymbol{a}} \pi_{\mathrm{jt}}^{\mathrm{dep}} Q_{\pi_{\mathrm{jt}}^{\mathrm{dep}}} = \sum_{\boldsymbol{a}} \pi_{\mathrm{jt}}^{\mathrm{ind}} Q_{\pi_{\mathrm{jt}}^{\mathrm{dep}}} = \mathbb{E}_{\boldsymbol{a}_t \sim \pi_{\mathrm{jt}}^{\mathrm{ind}}}[Q_{\pi_{\mathrm{jt}}^{\mathrm{dep}}}].$$

Thus, we can have:

$$\begin{aligned}
V_{\pi_{\mathrm{jt}}^{\mathrm{dep}}}(\mathrm{s}_t) &= \mathbb{E}_{\boldsymbol{a}_t \sim \pi_{\mathrm{jt}}^{\mathrm{ind}}}[Q_{\pi_{\mathrm{jt}}^{\mathrm{dep}}}(\mathrm{s}_t, \boldsymbol{a}_t)] \\
&= \mathbb{E}_{\boldsymbol{a}_t \sim \pi_{\mathrm{jt}}^{\mathrm{ind}}, s_{t+1} \sim P}[r(\mathrm{s}_t, \boldsymbol{a}_t) + \gamma V_{\pi_{\mathrm{jt}}^{\mathrm{dep}}}(s_{t+1})] \\
&= \mathbb{E}_{(\boldsymbol{a}_t, \boldsymbol{a}_{t+1}) \sim \pi_{\mathrm{jt}}^{\mathrm{ind}}, s_{t+1} \sim P}[r(\mathrm{s}_t, \boldsymbol{a}_t) + \gamma Q_{\pi_{\mathrm{jt}}^{\mathrm{dep}}}(\mathrm{s}_t, \boldsymbol{a}_t)] \\
&\vdots \\
&= \mathbb{E}_{\boldsymbol{a}_{t:\infty} \sim \pi_{\mathrm{jt}}^{\mathrm{ind}}, s_{t:\infty} \sim P}[r(\mathrm{s}_t, \boldsymbol{a}_t) + \gamma r(s_{t+1}, \boldsymbol{a}_{t+1}) + \cdots] \\
&= V_{\pi_{\mathrm{jt}}^{\mathrm{ind}}}(\mathrm{s}_t),
\end{aligned}$$

which concludes the proof. $\square$

## C  EXPERIMENT SETTINGS AND IMPLEMENTATION DETAILS

### C.1  MATRIX GAME

In the matrix game, we use a learning rate of $3 \times 10^{-4}$ for all algorithms. For FOP and MACPF, $\alpha$ decays from 1 to 0.5, with a decay rate of 0.999 per episode. For QMIX and QPLEX, $\epsilon$ decays from 1 to 0.01, with a decay rate of 0.999 per episode. The batch size used in the experiment is 64 for FOP, MACPF, QMIX, and QPLEX, and 32 for MAPPO as it is an on-policy learning algorithm. All critics and actors used in the experiments consist of one hidden layer of 64 units with ReLU non-linearity. For the Mixer network, QMIX and MACPF both use hypernetwork, except ELU non-linearity is used for QMIX and no non-linearity is used for MACPF. FOP and QPLEX both use attention network for their mixer network. The environment and model are implemented in Python. All models are built on PyTorch and are trained on a machine with 1 Nvidia GPU (RTX 1060) and 8 AMD CPU Cores.

### C.2  SMAC

In StarCraft II, for MACPF, we use a learning rate of $5 \times 10^{-4}$. The critic network and policy network of MACPF consist of three layers, a fully-connected layer with 64 units activated by ReLU, followed by a 64 bit GRU, and followed by another fully-connected layer. The policy correction network and critic correction network consist of two layers, one fully-connected layer with 64 units activated by

ELU, followed by another fully-connected layer. The target networks are updated after every 200 training episodes. The temperature parameters $\alpha$ and $\alpha_i$ are annealed from 0.5 to 0.05 over 200k time steps for all easy and hard maps and fixed as 0.001 for all super-hard maps. For QMIX, QPLEX, FOP, and MAPPO, we use their default setting of each map. The environment and model are implemented in Python. All models are built on PyTorch and are trained on a machine with 4 Nvidia GPUs (A100) and 224 Intel CPU Cores. For 3s5z_vs_3s6z, all models are built on PyTorch and are trained on a machine with 1 Nvidia GPU (RTX 2080 TI) and 16 Intel CPU Cores. Our implementation of MACPF is based on PyMARL (Samvelyan et al., 2019) with MIT license. It worth noting that, although we assume full observability for the rigorousness of proof, the trajectory of each agent is used to replace state $s$ for each agent as input to settle the partial observability in all SMAC experiments.

### C.3 MPE

In MPE (MIT license), we use the default settings of MAPPO. For QMIX, QPLEX, FOP, and MACPF, we use a learning rate of $5 \times 10^{-4}$. For FOP and MACPF, $\alpha$ decays from 0.5 to 0.05 over 50k time steps. For QMIX and QPLEX, $\epsilon$ decays from 1 to 0.05 over 50k time steps. The batch size used in the experiment is 64. All critics and actors used in the experiments consist of hidden layers of 64 units with ReLU non-linearity and 64 bit GRU. For the Mixer network, QMIX and MACPF both use hypernetwork, except ELU non-linearity is used for QMIX and no non-linearity is used for MACPF. FOP and QPLEX both use attention network for their mixer network. The environment and model are implemented in Python. All models are built on PyTorch and are trained on a machine with 1 Nvidia GPU (RTX 2080 TI) and 16 Intel CPU Cores. We also use the trajectory of each agent as input to settle the partial observability in all MPE experiments.

## D MORE EXPERIMENTS ON SMAC

### D.1 MORE MAPS

We additionally evaluate MACPF on more SMAC maps. The maps used here include six easy maps (8m, MMM, 3s_vs_3z, 3s_vs_4z, so_many_baneling, 1c3s5z), three hard maps (3s5z, 2c_vs_64zg, 3s_vs_5z) and three super-hard maps (3s5z_vs_3s6z, 27m_vs_30m, 6h_vs_8z). Results are shown in Figure 8. In general, MACPF matches or slightly outperforms the best performance of the baselines on all twelve maps.

### D.2 SUMMARY OF SMAC FINAL PERFORMANCE

In this section, we provide the summary of SMAC experiments in terms of final performance. All results are achieved by 2M training timesteps. As shown in Table 2, MACPF outperforms or at least matches the best performance of the baselines on all twelve maps.

## E MIXER SELECTION

As mentioned in Section 5.2, we use a hypernetwork without non-linearity as our mixer network, which differs from QMIX, QPLEX, and FOP. In QPLEX and FOP, weighted summation is used to reflect the relationship between $Q_{\text{jt}}$ and $Q_i$, where the weight is a function of both state and agent actions, such that the dependency among agents is implicitly considered. However, this implicit dependency may contradict our explicit dependency model in $Q_i^{\text{dep}}$ and decrease the performance of both $Q_{\text{jt}}^{\text{dep}}$ and $Q_{\text{jt}}^{\text{ind}}$.

Another choice is to use a hypernetwork with non-linearity to reflect the relationship between $Q_{\text{jt}}$ and $Q_i$, which is used in QMIX. However, due to the existence of the non-linearity unit, two joint actions with the same $Q_{\text{jt}}$ value may not be properly decomposed into two sets of $Q_i$ with the same sum. Thus, their joint probability may not be the same, and the dependency among agents is distorted.

Therefore, the only option left for MACPF is to use a hypernetwork without non-linearity, which is equivalent to weighted summation where the weight is just a function of state.

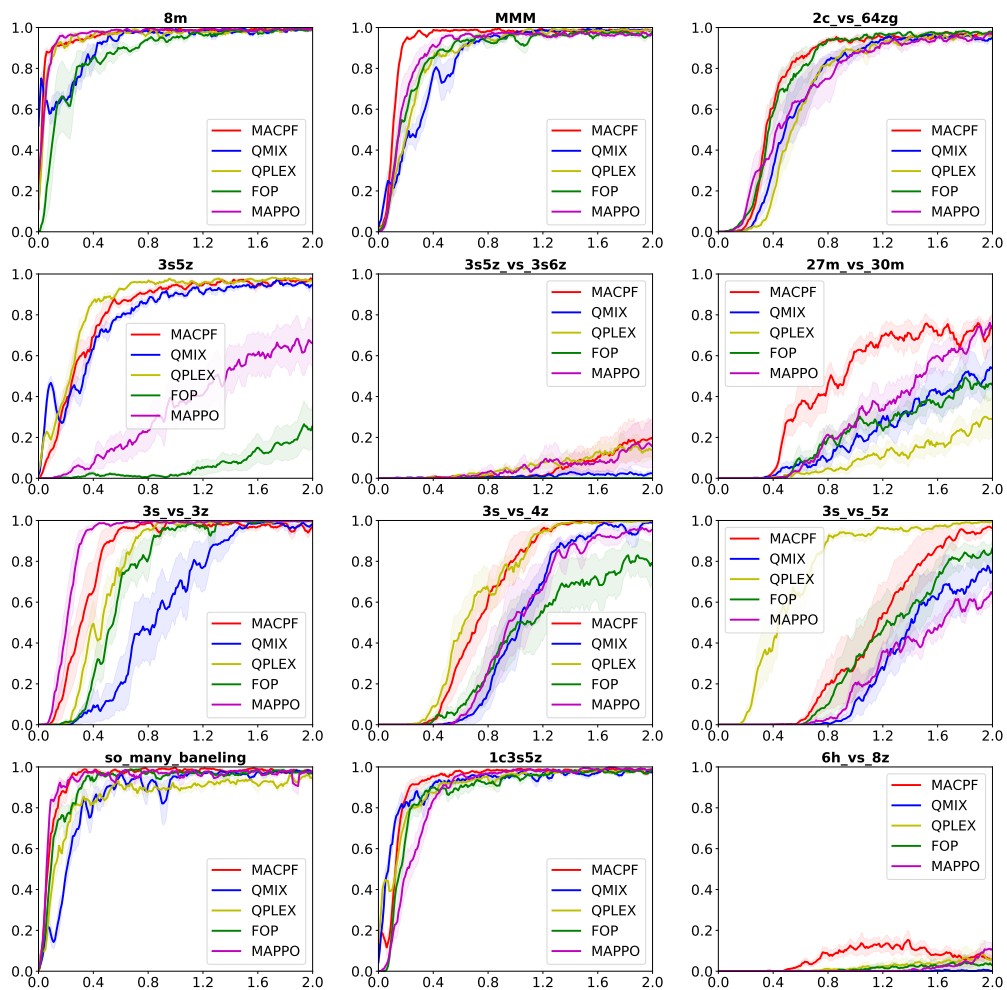

Figure 8: Learning curves of all the methods in twelve maps of SMAC, where the unit of x-axis is 1M timesteps and y-axis represents the win rate of each map.

As shown in Figure 9, MACPF_NONLINEAR and MACPF_ATT represent algorithms where all other aspects are the same as MACPF, except using a hypernetwork with non-linearity and a weighted summation with actions as input as their mixer networks, respectively. MACPF_NONLINEAR achieves similar performance as MACPF in the easy and hard maps, indicating that even distorted dependency can still benefit the learning. However, in the super-hard maps, MACPF outperforms MACPF_NONLINEAR, demonstrating the importance of accurate modeling of dependency among agents. MACPF_ATT is outperformed by both MACPF and MACPF_NONLINEAR by a large margin in all the maps, which verifies that the implicit dependency model in the mixer network of MACPF_ATT conflicts with the explicit dependency model in $Q_i^{\mathrm{dep}}$.

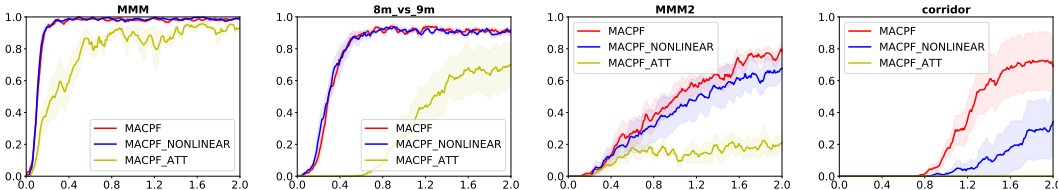

Figure 9: Ablation study of the mixer selection of MACPF on four maps of SMAC, including one easy map (MMM), one hard map (8m_vs_9m) and two super-hard maps (MMM2, corridor), where the unit of x-axis is 1M timesteps and y-axis represents the win rate of each map.

Table 2: Final performance on all SMAC maps. We bold all values within one standard deviation of the best mean performance for each map.

| Algorithms / Tasks | MACPF | QMIX | QPLEX | FOP | MAPPO |
|---|---|---|---|---|---|
| 8m (easy) | **0.994**±0.006 | 0.986±0.011 | 0.99±0.01 | 0.992±0.005 | **0.997**±0.003 |
| MMM (easy) | **0.984**±0.015 | **0.984**±0.01 | **0.985**±0.011 | **0.975**±0.017 | 0.962±0.035 |
| 2c_vs_64zg (hard) | **0.972**±0.031 | 0.946±0.013 | 0.954±0.031 | **0.976**±0.011 | 0.945±0.037 |
| 3s5z (hard) | **0.976**±0.008 | 0.955±0.017 | **0.969**±0.018 | 0.26±0.212 | 0.715±0.215 |
| 8m_vs_9m (hard) | **0.919**±0.045 | **0.916**±0.039 | 0.798±0.021 | 0.571±0.314 | 0.85±0.095 |
| 10m_vs_11m (hard) | **0.965**±0.035 | **0.939**±0.032 | **0.95**±0.016 | 0.545±0.265 | 0.774±0.106 |
| MMM2 (super-hard) | **0.788**±0.083 | **0.709**±0.162 | 0.224±0.231 | 0.506±0.144 | 0.679±0.054 |
| 3s5z_vs_3s6z (super-hard) | **0.209**±0.202 | **0.024**±0.031 | **0.135**±0.090 | 0.0±0.0 | **0.144**±0.175 |
| corridor (super-hard) | **0.691**±0.349 | 0.0±0.0 | 0.002±0.005 | 0.0±0.0 | **0.58**±0.184 |
| 27m_vs_30m (super-hard) | **0.726**±0.094 | 0.532±0.23 | 0.294±0.159 | 0.45±0.143 | **0.78**±0.095 |
| 8m_vs_9m (myopic) | **0.855**±0.069 | 0.675±0.127 | 0.716±0.075 | 0.338±0.329 | **0.81**±0.119 |
| 10m_vs_11m (myopic) | **0.888**±0.188 | **0.702**±0.129 | 0.664±0.089 | 0.384±0.372 | 0.514±0.253 |
| 3s_vs_3z (easy) | 0.974±0.019 | 0.988±0.014 | 0.994±0.004 | **0.999**±0.002 | **0.997**±0.003 |
| 3s_vs_4z (easy) | **0.995**±0.005 | 0.99±0.008 | **0.997**±0.003 | 0.789±0.22 | 0.957±0.022 |
| 3s_vs_5z (hard) | 0.959±0.033 | 0.759±0.153 | **0.992**±0.006 | 0.862±0.076 | 0.576±0.063 |
| so_many_baneling (easy) | **0.969**±0.019 | **0.974**±0.009 | 0.941±0.037 | **0.97**±0.025 | **0.979**±0.012 |
| 1c3s5z (easy) | **0.984**±0.006 | 0.98±0.013 | **0.985**±0.003 | **0.984**±0.005 | **0.989**±0.007 |
| 6h_vs_8z (super-hard) | **0.059**±0.038 | 0.001±0.002 | **0.059**±0.09 | 0.028±0.055 | **0.13**±0.074 |

## F   FUTURE WORK

One limitation of our work is the sequential decision-making process in training. Since the dependent local policy $\pi_i^{\text{dep}}(a_i \mid s, a_{<i})$ takes as input the joint action of all agents whose indices are smaller than agent $i$, agents have to make decisions one by one. This makes the whole decision process be $O(N)$. There is not much difference when $N$ is small. However, when $N$ is large, it slows down the training process. One approximate solution is to divide agents into groups, such that agents can make decisions group by group instead of one by one. However, such a mechanism may raise a new question about how to group agents, which will be considered in future work.

