# OpenReview forum: "More Centralized Training, Still Decentralized Execution: Multi-Agent Conditional Policy Factorization"
_ICLR.cc/2023/Conference — ICLR 2023 poster_

### Official Review · Reviewer_9QY5 · 2022-10-21

**Confidence:** 4
**Correctness:** 3
**Technical Novelty And Significance:** 3
**Empirical Novelty And Significance:** 3
**Recommendation:** 8

**Clarity, Quality, Novelty And Reproducibility:**

Clarity:Writing is clear

Quality:Looks good

Novelty:More interesting

Reproducibility:Not sure, need the author to open the source code


**Strength And Weaknesses:**

Strength.
* The paper is clearly written and well understood
* Looks interesting by introducing corrections in the central training
* The experimental section shows that the method proposed by the author works very well

Weaknesses.
* Equation (10) looks wrong because I don't see other additional constraints that make it mathematically reasonable since there is no way to ensure that the sum of \pi_i^{dep} is 1. Also Equation (10) is not consistent with the implementation in Fig 3.
* Equation (13) looks like it has many combinations possible, such as (Q - 10) + (c + 10), which means that its optimization space will become larger
* Equations (16) and (17) are not clear on what policy the choice of next action is based on
* Currently I am confused about the learning process of both ind and dep polices. In fig4, we can find that ind is slower than dep, but in fig7, the two polices are similar. Does this mean that there is no need to actually consider other agent's actions in smac? So why MACPF can outperform other strategies? Also is there a need for other experiments that consider agent?


**Summary Of The Paper:**

In order to better utilize the training paradigm of CTDE, the authors introduce dependency policy correction and dependency critic correction based on AC architecture to consider the action dependencies among the agents, and also design the corresponding network structure so that more accurate joint policy can be learned and thus more effective and better training results can be obtained.

**Summary Of The Review:**

The work is interesting, but I currently have some doubts and I will adjust my scores further after the rebuttal

---

> ### Author Response · Authors · 2022-11-14
> **Response to Reviewer 9QY5**
>
> Q1: Equation (10) looks wrong because I don't see other additional constraints that make it mathematically reasonable since there is no way to ensure that the sum of $\pi_i^{dep}$ is 1. Also Equation (10) is not consistent with the implementation in Fig 3.
>
> A1: Thanks for catching this. The combination of $\pi_i^{ind}$ and $b_i^{dep}$ is achieved by combining the logits of them and then softmax is used over the combined logit, such that the sum of $\pi_i^{dep}$ is always 1. We have added a corresponding footnote in our paper to better explain this point.
>
> Q2: Equation (13) looks like it has many combinations possible, such as (Q - 10) + (c + 10), which means that its optimization space will become larger
>
> A2: As stated in our paper, the optimization of Q and C is achieved in an alternating optimization method. We first optimize Q based on current C, and then C based on current Q. So, each time we will only have to deal with an optimization space of |Q| or |C| instead of |Q||C|, which makes this optimization process rather easy and stable.
>
> Q3: Equations (16) and (17) are not clear on what policy the choice of next action is based on
>
> A3: The next action is chosen by the current policy, and no target policy is used in our algorithm. We have made proper clarifications to our paper. Thanks for pointing this out.
>
> Q4: Currently I am confused about the learning process of both ind and dep polices. In fig4, we can find that ind is slower than dep, but in fig7, the two polices are similar. Does this mean that there is no need to actually consider other agent's actions in smac? So why MACPF can outperform other strategies? Also is there a need for other experiments that consider agent?
>
> A4: **but in fig7, the two polices are similar**, We assume you mean fig6 here? If so, we claim that, in SMAC, we do need to consider the dependency, which can be shown in Fig7. In Fig7, we use independent joint policy for both training and execution (In MACPF, we use dependent joint policy for training and independent joint policy for execution). We can see a large gap in performance in Fig7, which indicates the necessity of considering dependency in SMAC.
>
> The gap shown in fig4 is natural since the coordination pattern of Matrix Game is specifically designed to be very hard. Also, the learning process is relatively short in Matrix Game, so even a tiny gap becomes very obvious.
>
> In Matrix Game, we do need to consider other agents, as suggested by the payoff matrix. In SMAC, as stated above, the necessity of considering other agents is shown by the gap between MACPF and FOP, and also Fig7. In MPE, since there are only three agents involved, maybe the coordination pattern is relatively simple, and the necessity of considering other agents is relatively weak, so the performance gap between MACPF and other baselines is also relatively small.

---

> > ### Comment · Reviewer_9QY5 · 2022-11-18
> > **Reply To The Authors**
> >
> > Dear Authors,
> >
> > Thank you for your detailed replies. The authors have generally addressed my questions and feedback. But currently I still have some concerns about Q2 and Q4, please add more data or discuss further. If you can resolve my concerns, I will consider improving my score.

---

> > > ### Author Response · Authors · 2022-11-18
> > > **Reply to Reviewer 9QY5**
> > >
> > > Thanks for the feedback. We would like to elaborate more on these two questions.
> > >
> > > > I still have some concerns about Q2, please add more data or discuss further.
> > >
> > > The optimization space is indeed enlarged. However, since we are now modeling not only the utility function of each agent but also the effect of interaction between agents, this is the price we pay as we are modeling more items. To ease the problem of enlarged optimization space, we choose an alternating optimization method to optimize $Q_i$ and $C_i$ (also $\pi_i$ and $b_i$) to stabilize the training process. As suggested by the experiment results, it is worthwhile to pay such a price.
> > >
> > >
> > > Taking $Q_i$ and $C_i$ as an example. In each joint policy evaluation phase, we will first use (17) to get $Q_i^{new}$ based on $C_i^{old}$, as $Q_i$ considers only the utility of each agent, there is a possibility that we can not fit $Q_{jt}^{new}$ by solely changing $Q_i$. Therefore, we then use (16) to get $C_i^{new}$ based on $Q_i^{new}$. In this way, although we have to deal with one more variable, we can optimize them one by one to stabilize the overall optimization process.
> > >
> > >
> > >
> > > > I still have some concerns about Q4, please add more data or discuss further.
> > >
> > >
> > >
> > > To clarify this question, let us first revisit the experiment setup of Fig 6 and Fig 7.
> > >
> > >
> > >
> > > In Fig 6, we compare the performance of $\pi_{jt}^{dep}$ and $\pi_{jt}^{ind}$. This experiment aims to verify Theorem 2, where we state that there is always a $\pi_{jt}^{ind}$ that can achieve the same level of performance as $\pi_{jt}^{dep}$, and to show that our algorithm can indeed find this $\pi_{jt}^{ind}$. So the performance of $\pi_{jt}^{dep}$ and $\pi_{jt}^{ind}$ should match each other. But this is not because dependency is not important in SMAC since $\pi_{jt}^{ind}$ is also trained by trajectories collected by $\pi_{jt}^{dep}$, so it also enjoys the benefit of dependency. It simply indicates that the dependency is more about finding useful behavior, but once you find that behavior, you do not necessarily need the dependency to exploit it, as discussed in section 3.2
> > >
> > >
> > >
> > > As you mentioned in your question, in Fig 4, a similar experiment is done in the matrix game setting, but the performance of $\pi_{jt}^{dep}$ and $\pi_{jt}^{ind}$ does not match each other so well in the very beginning. We argue that this is because the matrix game is designed to be hard to converge for $\pi_{jt}^{ind}$ since those two optimal joint actions contradict each other, and $\pi_{jt}^{ind}$ can only select one of them, as suggested by the payoff matrix. Also, as we stated in our previous reply, the learning process of the matrix game is relatively short, so even a short period of mismatching becomes very obvious in that figure. Moreover, each data point in SMAC is an average of 32 evaluation episodes, and in the matrix game, it is just the result of one evaluation, which may also intensify the gap between $\pi_{jt}^{dep}$ and $\pi_{jt}^{ind}$.
> > >
> > >
> > >
> > > Regarding the importance of dependency in SMAC, it is shown in Fig 7, where the only difference between those two algorithms is whether $\pi_{jt}^{dep}$ is learned and used for exploration or not. As we can see in Fig 7, a huge gap between those two algorithms can be observed, so we claim that we have to consider dependency in SMAC, as the only difference between those two algorithms is whether we consider dependency or not. Furthermore, the necessity of considering dependency in SMAC has also been stated in [1], e.g., overkilling.
> > >
> > > [1] Samvelyan et al., The StarCraft Multi-Agent Challenge, 2019.

---

> > > > ### Comment · Reviewer_9QY5 · 2022-11-24
> > > > **Reply To The Authors**
> > > >
> > > > Dear Authors,
> > > >
> > > > Thank you for your detailed replies, the current ones solved my confusion. I adjusted my score to reflect my current attitude.
> > > >
> > > > But it is worth noting that, no matter whether the paper is finally accepted or not, please write clearly in the subsequent version to solve the concerns and doubts I mentioned.

---

> > > > > ### Author Response · Authors · 2022-11-24
> > > > > **Thanks**
> > > > >
> > > > > It is great our response has addressed your concern. We will definitely add a detailed discussion on these in the next version.
> > > > >
> > > > > Thanks for raising your score.

---

### Official Review · Reviewer_Eh9U · 2022-10-25

**Confidence:** 4
**Correctness:** 2
**Technical Novelty And Significance:** 3
**Empirical Novelty And Significance:** 3
**Recommendation:** 6

**Clarity, Quality, Novelty And Reproducibility:**

The proposed method looks novel by simultaneously learning both independent and dependent policies. The presentation is generally well-structured but can be improved with rigorousness.

**Strength And Weaknesses:**

**Strengths**
1. This paper studies the optimality of cooperative multi-agent reinforcement learning (MARL), which is an important problem.
2. Although the idea of conditional policy factorization is not novel, it is interesting to see this idea to be implemented with the deep neural network.
3. Empirical results on both a simple matrix game and more complex SMAC tasks show the effectiveness of the proposed method, compared to the baseline methods.

**Weaknesses**
1. The motivating example in Section 2.3 is confusing. From Section 2.2, FOP seems to have very limited expressiveness, the same capacity as a linear value factorization, e.g., VDN. This is because IGO factorization with the Boltzmann policy is equivalent to the linear value factorization. Therefore, FOP may not be a good candidate to show the limitations of independent policy factorization.
2. The idea of conditional policy factorization has been proposed in [1], which shall be cited and discussed.
3. Given the motivating example and the didactic matrix game, this paper seems to study multi-modality problems in cooperative MARL. This problem has been studied in [2], which should also be discussed or compared.
4. The reviewer tried QTRAN on the matrix game in Figure 4 and found it could quickly learn the optimal policy, which has different results from the paper and may invalidate the claim of this paper.
5.  Does the proposed method have a larger number of parameters than other methods, like FOP? If so, does this contribute to its outperformance?
6. How does the proposed method perform against baselines on all benchmark tasks in SMAC?
7. The paper is well-organized, but the presentation can be improved. The reviewer strongly suggests the authors use multi-agent MDPs to rewrite theorems and their proofs because Dec-POMDP with the infinite horizon is an undecidable problem. In addition, the approximation in Theorem 1 is quite informal.

[1] Dimitri Bertsekas. Multiagent rollout algorithms and reinforcement learning, 2019. https: //arxiv.org/abs/1910.00120
[2] Wei Fu, Chao Yu, Zelai Xu, Jiaqi Yang, Yi Wu. Revisiting Some Common Practices in Cooperative Multi-Agent Reinforcement Learning. ICML 2022.


**Summary Of The Paper:**

This paper studies the optimality issue of cooperative multi-agent reinforcement learning and claims that the independent policy factorization may lead to learning suboptimal policies. The authors propose a conditional policy factorization method, where an agent’s policy conditions on other agents (e.g., with lower indices). This paper claims that learning with such factorization theoretically leads to the optimal joint policy. It also shows that there exists an independent policy factorization that has the same value as conditional policy factorization, which enables decentralized execution. This paper empirically demonstrates the outperformance of the proposed method over baselines in a set of cooperative MARL tasks.

**Summary Of The Review:**

The proposed method is interesting and shows outperformance over baselines. However, it is still not quite intuitive to understand why the proposed method outperforms baselines (see weaknesses above). In addition, the motivation of this paper shall be strengthened.

---

> ### Author Response · Authors · 2022-11-14
> **Response to Reviewer Eh9U**
>
> Q1: FOP may not be a good candidate to show the limitations of independent policy factorization.
>
> A1: In FOP, attention model is used in mixer (like QPLEX), so it has better expressiveness than VDN. Also, to the best of our knowledge, FOP is the SOTA maximum-entropy MARL algorithm, therefore we claim that this is the only reasonable comparison we can make here.
>
> Q2&Q3: The idea of conditional policy factorization has been proposed in [1], which shall be cited and discussed. Given the motivating example and the didactic matrix game, this paper seems to study multi-modality problems in cooperative MARL. This problem has been studied in [2], which should also be discussed or compared.
>
> A2&A3: Thanks for pointing this out, we have added the corresponding citation in our paper. However, one should notice that both MA-rollout [1] and PG-AR [2] cannot be executed in a decentralized manner, therefore we will not include them as the baseline in our experiments.
>
>
> Q4: The reviewer tried QTRAN on the matrix game in Figure 4 and found it could quickly learn the optimal policy, which has different results from the paper and may invalidate the claim of this paper.
>
> A4: We follow the implementation of QTRAN in pymarl, and promise release our code in the supplementary materials once accepted. Please notice that QTRAN in our experiments do find optimal joint policy in some cases, but not all, so it is possible that by modifying some implementation details or changing random seeds, we can make QTRAN work in all cases. However, this will not invalidate our claim, as MACPF converge to the optimal policy in a more stable way, and also capture multimodal behavior. Also, the performance of QTRAN in SMAC tasks is in general not good, as shown in [3], so no matter how it performs in Matrix Game, it is still a rather theoretical work, and will not invalidate the necessity of our algorithm.
>
> Q5: Does the proposed method have a larger number of parameters than other methods, like FOP? If so, does this contribute to its outperformance?
>
> A5: No, in short, the number of parameters is in the same level of other baseline algorithms. In details, we use the exact same structure to process trajectories as FOP in both critic and actor. The only additional part in our algorithm is two MLP layers followed by ELU activation to process those actions from other agent, which is necessary for our algorithm. For mixer network, we use QMIX-style mixer without nonlinearity activation function, which is different from FOP in structure but still same level of parameterization. The necessity of using this specific mixer structure is fully discussed in Appendix E.
>
>
> Q6: How does the proposed method perform against baselines on all benchmark tasks in SMAC?
>
> A6: We have added three more maps. Due to the limitation of compute resources, we need more time to report the rest tasks, but we will keep updating them during the rebuttal session.
>
> Q7: The paper is well-organized, but the presentation can be improved. The reviewer strongly suggests the authors use multi-agent MDPs to rewrite theorems and their proofs because Dec-POMDP with the infinite horizon is an undecidable problem. In addition, the approximation in Theorem 1 is quite informal.
>
> A7: Thanks for bringing this to our attention. We have made corresponding modifications in our paper.
>
> [1] Multiagent Rollout Algorithms and Reinforcement Learning, Dimitri Bertsekas, https://arxiv.org/abs/1910.00120.
>
> [2] Revisiting Some Common Practices in Cooperative Multi-Agent Reinforcement Learning, Wei et al., ICML 2022.
>
> [3] QPLEX: Duplex Dueling Multi-Agent Q-Learning, Wang et al., ICLR 2020

---

> > ### Comment · Reviewer_Eh9U · 2022-11-27
> > **Thank you for the response**
> >
> > The response has partially addressed my concerns. I will increase my score to 6.

---

> > > ### Author Response · Authors · 2022-11-28
> > > **Thanks**
> > >
> > > Thanks for raising your score to 6.

---

> ### Author Response · Authors · 2022-11-18
> **Response to Reviewer Eh9U**
>
> Regarding your Q6, we added three more maps after the first reply. Now we have 18 maps in total. The win rate of MACFP is now higher than or equivalent to the best baseline in 16 out of 18 maps, while QMIX, QPLEX, MAPPO, and FOP are respectively 7/18, 8/18, 9/18, and 5/18. Please check the updated paper for details.

---

### Official Review · Reviewer_krAZ · 2022-10-25

**Confidence:** 2
**Correctness:** 3
**Technical Novelty And Significance:** 3
**Empirical Novelty And Significance:** 3
**Recommendation:** 6

**Clarity, Quality, Novelty And Reproducibility:**

**Clarity**

The paper is clearly written.

**Quality**

In general, the paper strikes me as technically sound, aside from the two points I mentioned before.

**Novelty**

As far as I know, the contributions of the paper are original, and improve over the state of the art.

**Reproducibility**

The results use a number of benchmark domains from the literature, and the paper provides in the appendix details on implementation. I did not check these carefully, but my impression is that the work is reproducible.

**Strength And Weaknesses:**

**Strengths**

The paper is very clearly written, and the ideas proposed are interesting and - to the extent of my knowledge - novel. The results also suggest that the proposed approach indeed pushes the performance of existing CTDE approaches.

**Weaknesses**

There are some steps in the proofs which I didn't fully grasp and which could be better discussed. Specifically,
1. I may have missed/misunderstood something, but it is not clear to me that the optimal policy factories as shown in (5), particularly in terms of the dependency on history. In light of this, I have some difficulty in making sense of Theorem 1, which seems to state that there is an optimal policy that verifies such factorization. Or is it stating that the algorithm converges to an optimal policy _over all policies that factorize as in (5)_?
2. Still on the point above (and, again, I may have missed/misunderstood something) in the proof of Theorem 1 it is not clear to me how the conclusion that the sequence of policies $\{\pi^k\}$ generated by the algorithm is improving (or, better said, not worsening) leads to the conclusion that the resulting policy is optimal overall joint policies. It seems to me that - in order for such a conclusion to hold - the limit must be unique from any initial policy, but it's not clear to me how such a conclusion follows from the proof...

**Summary Of The Paper:**

The paper addresses the problem of centralized training and decentralized execution (CTDE) in multiagent reinforcement learning. The paper contributes the MACPF algorithm (multi-agent conditional policy factorization), which relies on the idea that the joint policy given the history can be factorized conditioned on the actions of previous agents. In other words, in a $N$-agent Dec-POMDP,

$$\pi_{joint}(\boldsymbol a\mid\boldsymbol \tau)=\prod_{n=1}^N\pi_n(a_n\mid\tau_n,a_{<n}),$$

where $\tau_n$ is the history of agent $n$. Additionally, the paper observes that a dependent policy $\pi_n(a_n\mid\tau_n,a_{<n})$ can be written as the sum of an independent policy plus a correction term, i.e.,

$$\pi_n(a_n\mid\tau_n,a_{<n})=\pi_n(a_n\mid \tau_n)+b_n(a_n\mid \tau_n,a_{<n}),$$

where $b_n$ is the correction term. During the centralized training, the agents thus learn the dependent policy by learning the two components (the independent policy and the correction term). The former can then be used for decision-making during decentralized execution. The results portrayed in the paper indicate that the proposed approach indeed leads to improved performance over other competing approaches.

**Summary Of The Review:**

The paper proposes a novel approach for centralized training and decentralized execution in multiagent reinforcement learning. The proposed approach is novel and provides improvements over the state of the art. There are, however, a couple of technical issues that I did not fully understand and which could be better explained.

---

> ### Author Response · Authors · 2022-11-14
> **Response to Reviewer krAZ**
>
> Thanks for acknowledging our contribution.
>
> Q1: I have some difficulty in making sense of Theorem 1, which seems to state that there is an optimal policy that verifies such factorization. Or is it stating that the algorithm converges to an optimal policy over all policies that factorize as in (5)?
>
> A1: By Theorem 1, we are stating that this algorithm will converge to an optimal joint policy within the policy class induced by (5).
>
> Q2: It seems to me that - in order for such a conclusion to hold - the limit must be unique from any initial policy, but it's not clear to me how such a conclusion follows from the proof...
>
> A2: No, there can be multiple optimal joint policies. In Theorem 1, the optimality is not a direct result of convergence (we assume that is what you mean. If every sequence converges and that limit is unique, then we have optimality). To get optimality, we will first have $E_{\pi_{jt}^*}[Q^{\pi_{jt}^*} - \log \pi_{jt}^*] \geq V_{jt}^{\pi_{jt}}$, following the same proof procedure of (26). Then, by repeatedly expanding $Q^{\pi_{jt}^*}$ and plugging the above inequality, we can have $Q^{\pi_{jt}^*} \geq Q^{\pi_{jt}}$. You can notice that the same proof technique is also used in [1] and [2].
>
>
> [1] Soft actor-critic: Off-policy maximum entropy deep reinforcement learning with a stochastic actor, Haarnoja et al., ICML 2018.
>
> [2] Fop: Factorizing Optimal Joint Policy of Maximum-Entropy Multi-Agent Reinforcement Learning, Zhang et al., ICML 2021.

---

> > ### Comment · Reviewer_krAZ · 2022-11-24
> > **Thank you for your response**
> >
> > I thank the reviewers for their response. I am happy about the response to Q1. As for Q2, I may be missing something, but it is not clear to me how you move from concluding that $Q^{\pi^*_{\mathrm{jt}}}$ is optimal across all $\pi_{\mathrm{jt}}$, and not only in the sequence {$\pi_{\mathrm{jt}}^k$}.

---

> > > ### Author Response · Authors · 2022-11-24
> > > **Response to Reviewer krAZ**
> > >
> > > Thanks for your response again. We would like to elaborate more on this to clarify your concern.
> > >
> > > When the sequence {$\pi_{jt}^{k}$} converges, there must be $\pi_{jt}^{k} = \pi_{jt}^{k+1}$, we can mark it as $\pi_{jt}^*$. As $\pi_{jt}^*$ is obtained via the optimization problem defined as Eq. (22), $\pi_{jt}^*$ must be the minimizer of this problem among all $\pi_{jt} \in \Pi_1 \times \cdots \times \Pi_N$. One may notice that Eq. (22) is defined for each $\pi_i$, however, due to our specific $Q_{jt}$ factorization, the minimizer of each $\pi_i$ can jointly form the optimizer of $\pi_{jt}$.
> > >
> > > Therefore, we can have the following equation which holds for all $\pi_{jt} \in \Pi_1 \times \cdots \times \Pi_N$:
> > >
> > > $J_{\pi_{jt}^*}(\pi_{jt}^*(\cdot|s)) \leq J_{\pi_{jt}^*}(\pi_{jt}(\cdot | s)), \forall \pi_{jt} \neq \pi_{jt}^*.$
> > >
> > > Then, we can expand KL and follow the same procedure in Eq. (23), (24) and (25) to get the following inequality, which still holds for all $\pi_{jt} \in \Pi_1 \times \cdots \times \Pi_N$:
> > >
> > > $E_{\pi_{jt}^*}[Q^{\pi_{jt}^*} - \log \pi_{jt}^*] \geq V_{jt}^{\pi_{jt}}$
> > >
> > >
> > > We can then repeatedly expand $Q^{\pi_{jt}^*}$ and plug the above inequality, and get the following inequality, which also holds for all $\pi_{jt} \in \Pi_1 \times \cdots \times \Pi_N$:
> > >
> > >
> > > $Q^{\pi_{jt}^*} \geq Q^{\pi_{jt}}.$
> > >
> > >
> > >
> > > This concludes the proof of Theorem 1. Please notice that the optimization problem is defined over all $\pi_{jt} \in \Pi_1 \times \cdots \times \Pi_N$. Although we may have different convergence point from different {$\pi_{jt}^{k}$} sequence, but the property of convergence is enough for us to guarantee the converged $\pi_{jt}^{*}$ being optimal among all $\pi_{jt}$ and also other inequalities mentioned above to be held, as the sequence {$\pi_{jt}^{k}$} is obtained via this minimization problem (22). Please refer to Appendix A for details.

---

### Official Review · Reviewer_FBdE · 2022-11-05

**Confidence:** 3
**Correctness:** 4
**Technical Novelty And Significance:** 2
**Empirical Novelty And Significance:** 2
**Recommendation:** 5

**Clarity, Quality, Novelty And Reproducibility:**

They describe clearly the proposed method, and I think that the paper has the good quality. If the authors will give the hyperparameter in their experiments, it is helfpul for reproducibility.


**Strength And Weaknesses:**

- Streagth
Actually, learn the joint-policy, it can gaurantee the optimal policy in joint-aciton space.
Ouput the decentrazlied individual policies distilled from the dependent policies

- Weakness
  Complex network structure, do exist dependent and individual networks ( $Q^{dep}_i, Q^{int}_i$ )
  To learn the individual policies from joint dependency policies, they propose the naive method


**Summary Of The Paper:**

This paper deals with the problem of decentralized policy in MARL when using the factorizing value method, especially FOP. To solve this problem, they propose **multi-agent conditined policy factorization** (MACPF) that incorporate dependency between the agents.
Due to the fact that the proposed method learns the joint policy, it can guarantee to the optimal joint policy. They
They give the suitable network structure for the condition policy, also show the proper experiment results both in toy-example and complex task, SMAC.


**Summary Of The Review:**

Learning the coordinated strategies in multi-agent is challenging because of partial-observability.
The author gives the intuition about what the problem is in the previous value-factorized method. To learn the opitmal policy, the depedency between agents should be considered, but it can break the decentralied execution in the test time. To keep the decentralized execution, having the factorized policies, they build the two seperate networks, dependent and individual networks. The proposed network structure is not simple, but they show the good performance by using the valued-based method when comparing strong baselines.

---

> ### Author Response · Authors · 2022-11-14
> **Response to Reviewer FBdE**
>
> Q1: Complex network structure, do exist dependent and individual networks
>
> A1: Since we are using two different policies in our algorithm, at least two networks are needed for them. From this perspective, our algorithm is as simple as possible in this framework. Also, although our algorithm is slightly more complex than those algorithms with only one network, such complexity is supported by theory and has proven to be useful in our experiments. So, we argue that the complexity of our work is not a flaw.
>
> Q2: To learn the individual policies from joint dependency policies, they propose the naive method.
>
> A2: First, we do not learn individual policies from joint policy, we learn individual independent policies from their individual dependent policies by introducing correlation between them, and to the best of our knowledge, this is the first work using such a method to achieve this goal. Second, this is not a naive method since it is supported by both theoretical results and empirical results, so we prefer to call it "simple but effective." Lastly, as suggested by other reviewers, there is a certain level of novelty in our work, and it has not been done before, so maybe it is not that "naive."

---

### Author Response · Authors · 2022-11-24
**General response to reviewers**



To all reviewers:



Thanks a lot for your valuable comments. We have made proper modifications according to your suggestions. All modifications are marked in blue in our revised paper. To save you some time, we briefly summarize all changes here as follows:





1. Related Work



* Citations of MA-rollout and PG-AR have been added and discussed in the introduction section of the revised paper.



2. Problem definition



*  We have changed our problem definition to multi-agent MDP in the revised paper and made proper modifications in both the proof and framework sections.



3. Experiments



* We have added results on additional 6 SMAC maps. Now we have in total 18 maps and the win rate of MACFP is now higher than or equivalent to the best baseline in 16 out of 18 maps, while QMIX, QPLEX, MAPPO, and FOP are respectively 7/18, 8/18, 9/18, and 5/18. Detailed results about these new maps have been added in Appendix D of the revised paper.



4. Details



* We have clarified how $a'$ is selected in (16) and (17) in section 3.3 of the revised paper.



* We have added details about how to get $\pi_i^{dep}$ using $\pi_i^{ind}$ and $b_i^{dep}$ in section 3.3 of the revised paper.



**We hope that our responses have addressed all the questions and concerns. If so, please consider raising your initial score to reflect on it. If we missed anything, please let us know. We are always willing to answer any of your questions.**

---

### Decision · Program_Chairs · 2023-01-20

**Decision:**

Accept: poster

**Justification For Why Not Higher Score:**

In the experiments, the improvement is mostly consistent across many different domains, but super hard tasks are still super hard for MACPF.

**Justification For Why Not Lower Score:**

The method is technically sound and principled, and certainly advances the state of the art.

**Metareview: Summary, Strengths And Weaknesses:**

This paper addresses the problem of centralized training and decentralized execution (CTDE) in multi-agent RL. The proposed algorithm, MACPF (multi-agent conditional policy factorization), uses the fact that the joint policy can be factorized by individual policies dependent on other agents (essentially chain rule for joint probabilities), and these individual policies could be decomposed into the individual/independent policy term and a correction term. These terms are learned during centralized training. The results show significant improvement. Overall, this is a novel and convincing idea to improve CTDE in MARL.

All the questions and issues were clearly resolved by author response. This paper would make a solid contribution.

Most of the figures are almost illegible (small fonts, cramped plots, etc). Please revise the figures in the final version.

**Note From Pc:**

if the above contains the word "oral" or "spotlight" please see: "oral" presentation means -> notable-top-5% and "spotlight" means -> notable-top-25%. As stated in our emails, we are disassociating presentation type from AC recommendations

**Summary Of Ac-Reviewer Meeting:**

All the reviewers appreciated the novelty and the technical soundness of the approach. The empirical results were quite impressive.

All the points raised by the reviewers were clearly resolved by author response.